# ROC-N-REROLL: How verifier imperfection affects test-time scaling

**Florian E. Dorner**[1,2,3,4]**, Yatong Chen**[*3,4]**, André F. Cruz**[*3,4]**, and Fanny Yang**[1]

[1]ETH Zürich, [2]Max Planck ETH Center for Learning Systems,
[3]Max Planck Institute for Intelligent Systems, Tübingen, [4]Tübingen AI Center

## ABSTRACT

Test-time scaling aims to improve language model performance by leveraging additional compute during inference. Many works have empirically studied techniques such as Best-of-N (BoN) and Rejection Sampling (RS) that make use of a verifier to enable test-time scaling. However, to date there is little theoretical understanding of how verifier *imperfection* affects performance — a gap we address in this work. Specifically, we prove that the instance-level accuracy of these methods is precisely characterized by the geometry of the verifier's ROC curve. Our theory has two important takeaways, confirmed by experiments with Qwen and LLama models on GSM8K and MATH500. First, for any query with a concave verifier ROC curve, RS outperforms BoN for fixed compute, while both methods converge to the same accuracy in the infinite-compute limit. Second, it is generally impossible to predict the high-compute performance of either method based on observations in the low-compute regime.

## 1 INTRODUCTION

Just as further scaling up large language model (LLM) pre-training started to show diminishing returns, OpenAI released o1, vastly improving upon the state-of-the-art on many challenging benchmarks (OpenAI, 2024). Instead of spending more compute on pre-training, o1 was the first flagship LLM to prominently improve performance by spending additional compute at *test-time*. Since then, interest in *test-time scaling* has exploded (Muennighoff et al., 2025; Guo et al., 2025; Kimi et al., 2025; Qu et al., 2025; Aggarwal and Welleck, 2025; Zaremba et al., 2025; Kavukcuoglu, 2025).

There are two broad approaches to test-time scaling: resampling and "reasoning". Both approaches typically use a *verifier* — a scoring mechanism that evaluates the quality or correctness of an LLM's outputs — but at different stages of the pipeline. Resampling methods employ a verifier at test-time to filter or rank candidate responses after they are generated (Cobbe et al., 2021). In contrast, reasoning methods employ a verifier to modify how the LLM generates outputs, usually increasing output quality at the cost of increased response length. Most prominently, the verifier can be used as a reward for post-training with reinforcement learning (RL) (Guo et al., 2025).

In practice, both test-time scaling approaches have primarily been successful in domains where a *reliable* oracle verifier can be implemented — e.g., coding using unit tests and math using ground-truth numerical solutions. As such, previous theoretical analysis has focused on the scaling behavior of *pass@N*, the probability that at least one of the $N$ candidate responses is correct (Brown et al., 2024; Schaeffer et al., 2025). In most domains, however, access to a perfectly accurate verifier is not realistic. errors may slip through: insecure code can pass static tests (Zhou et al., 2024), and flawed reasoning can arrive at the correct numerical answer (Petrov et al., 2025). More broadly, there has been an increasing interest in using *another language model* as a verifier (Huang et al., 2025a; Song et al., 2025), an approach that can be applied to any domain, but has been shown to have far from perfect accuracy (Bavaresco et al., 2025; Dorner et al., 2025).

Despite growing interest in verifier-based test-time scaling, the relationship between scaling behavior and the properties of imperfect verifiers remains poorly understood. This work addresses this

---

[*]Equal contribution.   Code: https://github.com/socialfoundations/roc-n-reroll

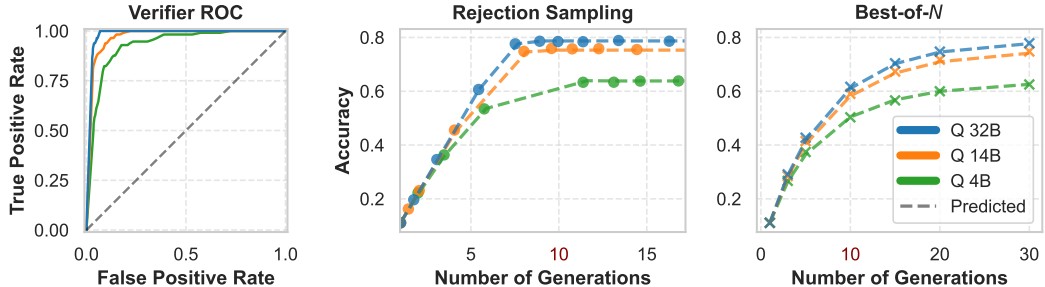

Figure 1: Empirical performance (markers) of RS (middle) and BoN (right) on GSM8K test question 58, overlaid with theoretical predictions (lines). Different verifiers scale similarly at first, but then diverge. RS matches BoN accuracy, using less average compute. Generator: `Qwen3-1.7B`.

gap. We analyze two simple resampling methods for test-time scaling: *Rejection Sampling* (RS), which resamples answers until the verifier score exceeds a predetermined threshold, and *Best-of-N* (BoN), which samples $N$ answers and returns the highest-scoring one. We provide a series of theoretical and empirical results that connect the performance and compute costs of both methods to a classical concept in machine learning: the *Receiver Operating Characteristic* (ROC) curve (Peterson et al., 1954). For a fixed query, the verifier's ROC curve encodes all possible trade-offs between two types of errors: false negatives – rejecting correct answers, and false positives – accepting incorrect answers. Specifically, our contributions are as follows:

- We prove that for a given query, the accuracy of both BoN and RS only depends on the generator and verifier via the generator's initial accuracy and the verifier's ROC curve (Propositions 1 and 4). Accuracy is agnostic to implementation details beyond that.

- For concave ROC curves, we prove that RS outperforms BoN for fixed compute (Proposition 5), but converges to the same accuracy as compute increases (Proposition 2 and Theorem 1). Further, extrapolation based on early scaling is impossible in both cases – as seen in Figure 1, performance at high numbers of test-time samples can vary widely between verifiers, even if accuracy is identical at small numbers of samples (Propositions 3 and 6).

- Using Qwen3 and LLama models as verifiers, we validate the accuracy of our per-instance performance predictions on a subset of GSM8K questions (see Figure 1, Figure 3). We also confirm our high-level takeaways on the full GSM8K and MATH500 datasets (Figure 4a).

The rest of this paper is structured as follows: We begin by discussing related work (Section 2). We then present our formal setup (Section 3), followed by our theoretical results for RS (Section 4) and BoN (Section 5). Lastly, we conclude by presenting our experimental results (Section 6).

## 2 RELATED WORK

Test-time scaling methods can be broadly divided into two categories: *resampling* methods that aggregate multiple LLM outputs - and *"reasoning"* methods that modify an LLM to elicit longer responses with "human-like" reasoning steps (see Appendix A.1 for additional discussion and Zhang et al. (2025) for a survey). Despite the recent popularity of reasoning methods, resampling is still prominently applied in industry releases: While OpenAI reports majority voting results (OpenAI, 2024), and Anthropic's Claude 4 uses BoN in its "high compute" mode (Anthropic, 2025), DeepMind's AlphaCode (Li et al., 2022) uses test cases to filter generated code, and AlphaEvolve (Novikov et al., 2025) uses numeric feedback to iterate on and refine proposed solutions.

**Rejection Sampling** RS is routinely used to create synthetic data for model training (Zelikman et al., 2024; Yehudai et al., 2024; Uesato et al., 2022; Zhu et al., 2023; Xiong et al., 2025; Yuan et al., 2023; Dorner et al., 2023), mostly in settings with a single canonical verifier. However, as a method for test-time scaling, RS has not received much attention in the literature to date. This is likely due to two practical disadvantages: Unlike for BoN, the sampling budget can only be controlled indirectly via choosing a decision threshold, and parallelization is difficult. Still, Ziegler

et al. (2022) use RS for safety filtering, while Chen et al. (2024) study the interplay between majority voting and LLM-based answer filtering. Most similar to our work, Song et al. (2025) empirically investigate the performance of RS when the generator and the verifier are based on the same LLM, while Stroebl et al. (2024) find that scaling up a variant of rejection sampling for a fixed binary unit-test based verifier has limited benefits. Our work precisely characterizes the compute-scaling of RS performance, based on the ROC curve, and show that RS (partially) compensates for its practical disadvantages via improved performance compared to BoN at a fixed compute level.

**Best-of-N** For BoN, theoretical work has analyzed the case of perfect verifiers, in which case BoN performance is equivalent to *pass@N*: Brown et al. (2024) estimate pass@N scaling based on a per instance closed-form formula for expected accuracy and find aggregate performance to approximately follow a power law. Meanwhile, Schaeffer et al. (2025) point out that the closed-form solution does not imply power law scaling per instance. The authors reconcile this by hypothesizing that the observed aggregate power-law scaling is caused by a heavy tail in the distribution of instance difficulties. That said, BoN can only achieve pass@N performance if the verifier is perfect, which is not realistic in most practical applications. While Chen et al. (2025) relax the perfect-verifier assumption to better-than-chance pairwise comparisons, their results require errors to be independent across repetitions, which allows for arbitrarily high verifier accuracy via boosting.

BoN with imperfect proxy scores $f(X)$ has attracted substantial empirical interest (Cobbe et al., 2021; Gao et al., 2023; Coste et al., 2024). However, most theoretical work on BoN has focused on how the number of samples $N$ affects the answer quality as measured by the *verifier score $f(X)$* (Beirami et al., 2025; Gui et al., 2024). Instead, our work focuses on how the *ground-truth performance* scales for unreliable verifiers. Most related to our work, a recent paper by Huang et al. (2025b) provides bounds on BoN performance, based on the mean squared error (MSE) between the score $f(X)$ and the ground truth reward $y(X)$. Rather than providing bounds, our work uses the ROC curve to *fully* characterize BoN performance in the context of binary ground truth rewards.

## 3 FORMAL SETUP

Throughout the rest of this work, we consider a fixed query $q$ and a generative model $g_{\text{base}}$ (the *generator*) that produces responses $X \in \mathcal{X}$. Let $P_{g_{\text{base}}}$ denote the distribution over $\mathcal{X}$ induced by sampling from $g_{\text{base}}$ (conditioned on the query $q$). We assume an unknown ground-truth labeling function $y : \mathcal{X} \mapsto \{0, 1\}$, where $y(X) = 1$ iff $X$ is a correct answer to the query $q$. In addition, we have access to a verifier score $f : \mathcal{X} \mapsto [0, 1]$ that is (supposedly) correlated with $y$. For example, this might be another LLM's assessment of the correctness of the answer $X$ to the query $q$. Based on $f$, we can then define a binary classifier $h^\tau : \mathcal{X} \mapsto \{0, 1\}$ by thresholding $h^\tau(X) = \mathbb{I}[f(X) \geq \tau]$, where $\mathbb{I}$ is the indicator function.

We now define key performance quantities of the generative model $g_{\text{base}}$ and a binary classifier $h$:

- $\pi := \text{ACC}(g_{\text{base}}) = P_{g_{\text{base}}}[y(X) = 1]$ : the accuracy of the generative model $g_{\text{base}}$
- $\text{T}(g_{\text{base}}, h) = P_{g_{\text{base}}}[h(X) = 1 | y(X) = 1]$: the true positive rate (TPR) of the classifier $h$
- $\text{F}(g_{\text{base}}, h) = P_{g_{\text{base}}}[h(X) = 1 | y(X) = 0]$: the false positive rate (FPR) of the classifier $h$

Further, for a fixed verifier $f$, we define $\mathcal{H}(f)$ as the set of all classifiers $h^\tau$, where $\tau \in [0, 1]$. We then refer to the classifier that maximizes true positive rate for a given false positive rate $F$ as

$$h_{\text{F}} := \underset{h \in \mathcal{H}(f):\ \text{F}(g_{\text{base}}, h) \leq \text{F}}{\arg\max} \text{T}(g_{\text{base}}, h). \tag{1}$$

With this, we can formally describe the ROC curve of a verifier $f$:

**Definition 1.** *(ROC Curve) Given a fixed generator $g_{base}$ and a verifier $f$, the* ROC curve *of $f$ is the function $T : [0, 1] \to [0, 1]$ defined by: $T(F) := T(g_{base}, h_F) = \max \{T(g_{base}, h) : h \in \mathcal{H}(f),\ F(g_{base}, h) \leq F\}$.*

In words, the ROC curve describes the true positive rate of all optimal classifiers $h_{\text{F}}$ and thus the Pareto optimal tradeoffs between $\text{F}(g_{\text{base}}, h)$ and $\text{T}(g_{\text{base}}, h)$. Note that ROC curves $\text{T}(F)$ are (non-strictly) increasing in F. For additional context on ROC curves, see Appendix A.2.

### 3.1 Two Methods for Test-time-scaling

In the following sections, we fix a generator $g$ and verifier $f$, and analyze the scaling of rejection sampling and Best-of-N. Both methods induce a new generative distribution over outputs $X$.

**Rejection Sampling (RS):**   We repeatedly draw outputs $X \sim P_{g_{\text{base}}}$ and apply a classifier $h_{\text{F}} \in \mathcal{H}(f)$ to each sample $X$ to determine acceptance. The process halts and returns the first sample $X$ such that $h_{\text{F}}(X) = 1$. RS thus defines a new generative process $g^{h_{\text{F}}}$ corresponding to the conditional distribution of $X$ given $h_{\text{F}}(X) = 1$, i.e.,

$$P_{g^{h_{\text{F}}}}[X = x] := P_{g_{\text{base}}}[X = x | h_{\text{F}}(X) = 1]. \tag{2}$$

Decreasing the false positive rate F or increasing the threshold $\tau$ causes RS outputs $X$ to have higher verifier scores $f(X)$ at the cost of longer running times due to fewer outputs $X$ being accepted.

**Best-of-N (BoN):**   We draw $N$ independent samples $X_1, \cdots, X_N \sim P_{g_{\text{base}}}$ from the generator and return the one with the highest score under $f$, i.e., $\arg\max_{\{X_i\}_{i \in [N]}} f(X_i)$ with ties broken randomly. BoN induces a new generator $g_N^f$ with outputs distributed as

$$P_{g_N^f}[X = x] := P_{g_{\text{base}}}^{\otimes N} \big[ \arg\max_{X' \in \{X_i\}_{i \in [N]}} f(X') = x \big], \tag{3}$$

where $P_{g_{\text{base}}}^{\otimes N}$ denotes the joint distribution of $N$ independently sampled $X_i \sim P_{g_{\text{base}}}$. The running time of BoN can straightforwardly be adapted by increasing $N$, improving the average score $f(X)$. For a summary of the notation introduced in this section and used throughout the paper, see Table 2.

## 4 Rejection Sampling

In the following section, we characterize how the ROC curve of a fixed verifier $f$ affects the test-time scaling of the RS in terms of two key quantities: The compute cost of RS, and the accuracy $\text{ACC}(g^{h_{\text{F}}})$ of the RS distribution $P_{g^{h_{\text{F}}}}$ defined in Equation (2) for $h_{\text{F}} \in \mathcal{H}(f)$. We begin by deriving how the accuracy and compute cost of the generative process $g^{h_{\text{F}}}$ vary with the false positive rate F.

**Compute cost**   Let $N(\text{F})$ denote the number of samples $X$ drawn from the base distribution $P_{g_{\text{base}}}$ until $h_{\text{F}}$ in Equation (1) accepts, i.e., $h_{\text{F}}(X) = 1$. Then the average compute cost $C(\text{F})$ of RS – measured in terms of the number of samples and verifications[1] – corresponds to $\mathbb{E}[N(\text{F})]$. Since $N(\text{F})$ follows a geometric distribution with success probability $P_{g_{\text{base}}}[h_{\text{F}}(X) = 1]$, we have

$$C(\text{F}) := \mathbb{E}[N(\text{F})] = \frac{1}{P_{g_{\text{base}}}[h_{\text{F}}(X) = 1]} = \frac{1}{\text{T}(\text{F}) \cdot \pi + \text{F} \cdot (1 - \pi)} \tag{4}$$

where $\pi = \text{ACC}(g_{\text{base}})$ is the accuracy of the generator $g_{\text{base}}$. Note that $C(\text{F}) = \frac{1}{\pi}$ for a perfect classifier $h_{\text{F}} = y$, while $C(\text{F}) \to \infty$ when the probability of $h_{\text{F}}$ accepting an output $x$ tends to zero.

**Accuracy**   RS is equivalent to the precision or positive predictive value $P_{g_{\text{base}}}[y(X) = 1 | h_{\text{F}}(X) = 1]$ of the classifier $h_{\text{F}}$. As observed by Song et al. (2025), modifying the decision threshold $\tau$ and thus F and $h_{\text{F}}$ induces different accuracies for the output distribution $\text{ACC}(g^{h_{\text{F}}})$. Combining Equation (1) and Definition 1, as well as the Bayes rule, we can write

$$\bar{A}_f(\text{F}) := \text{ACC}(g^{h_{\text{F}}}) = \frac{\text{T}(\text{F}) \cdot \pi}{\text{T}(\text{F}) \cdot \pi + \text{F} \cdot (1 - \pi)}. \tag{5}$$

In particular, $\bar{A}_f(\text{F}) = \pi(g)$ when $h_{\text{F}}(X)$ is independent of $y(X)$, while for $\pi > 0$ and a perfect classifier $h_{\text{F}} = y$, we get $\bar{A}_f(\text{F}) = 1$. Because $\text{T}(\text{F})$ increases in F, $C(\text{F})$ defined in Equation (4) decreases strictly. Thus, the function $C(\text{F})$ has an inverse $\text{F}(C)$. With this, a change of variables in Equation (5) yields an expression for accuracy directly in terms of the expected compute $C$, i.e.,

$$A_f(C) := \bar{A}_f(\text{F}(C)) = \frac{\text{T}(\text{F}(C)) \cdot \pi}{\text{T}(\text{F}(C)) \cdot \pi + \text{F}(C) \cdot (1 - \pi)} = C \cdot \pi \cdot \text{T}(\text{F}(C)) \tag{6}$$

---

[1]Note, that due to Wald's equation, the expected number of tokens generated by an LLM (and an LLM-based verifier) during RS is proportional to the expected number of samples. For details, consider Appendix B.1.

Throughout most of this section, we will drop the subscripts and write $A(C)$ and $\bar{A}(F)$ respectively. In the next proposition, we derive the slope of the compute-performance curve as a function of the slope of the ROC curve T(F) and show that concave ROC curves imply monotonous scaling for RS.

**Proposition 1.** *Let $f$ be a verifier and $T : [0,1] \mapsto [0,1]$ be the ROC curve of $f$. If the derivative $T'(F)$ exists at F, the derivative of the accuracy-compute curve at $C(F)$ is given by*

$$\frac{dA(C)}{dC}\bigg|_{C=C(F)} = \pi \frac{(1-\pi)\left(T(F) - FT'(F)\right)}{1 + \pi T'(F) - \pi}.$$

*For (strictly) concave ROC curves, $\dfrac{dA(C)}{dC}\bigg|_{C=C(F)}$ is (strictly) positive whenever $T'(F)$ exists.*

Proposition 1 is proven in Appendix C.1. Beyond monotonous scaling for concave ROC curves, it implies that when the ROC curve T(F) is piecewise linear, so is the accuracy $A(C)$.

## 4.1 LOW-COMPUTE REGIME

We now analyze the performance of RS in the *low-compute regime*, which corresponds to using a classifier with high FPR (e.g., F $\approx$ 1) that accepts almost all outputs without much filtering. At the extreme of F = 1 at the top-right corner of the ROC curve, the generative process $g^{h_1}$ induced by the classifier $h_1$ samples exactly once per accepted output, minimizing compute. As we slightly tighten the classifier by decreasing F from 1, compute increases and performance improves at the rate of

$$\frac{dA(C)}{dC}\bigg|_{C=1} = \pi \frac{(\pi-1)(1 - T'(1))}{\pi - 1 - \pi T'(1)}. \tag{7}$$

In particular, for the minimal possible slope $T'(1) = 0$, accuracy initially grows with compute at a rate of $\pi$. On the other extreme, when $T'(1) = 1$, there is no improvement with increased compute.

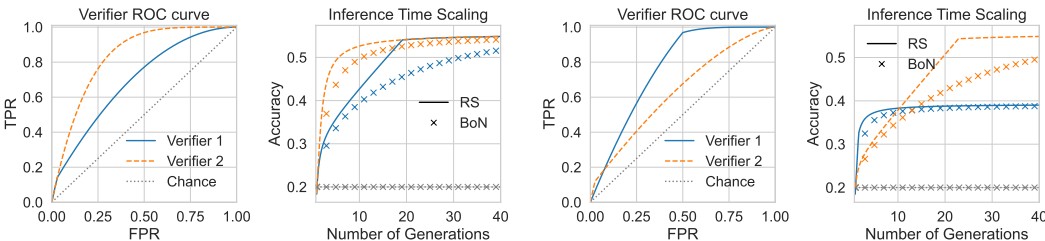

(a) Different small-, same large-scale performance      (b) Early scaling reverses at large scale

Figure 2: Performance of RS (line) and BoN (scatter) with different verifiers (synthetic data).

Figure 2a illustrates how the top-right corner determines early scaling: We plot two ROC curves that behave differently in the top-right corner, but the same in the bottom-left corner. As predicted by Equation (7), RS scales more quickly when the ROC curve is more "flat" near the top-right corner. Interestingly, the similarity of the ROC curves near the origin appears to coincide with diminishing performance differences in the high-compute limit. In the next section, we prove this observation.

## 4.2 HIGH-COMPUTE REGIME

We now characterize the performance of RS in the *high-compute* regime, which corresponds to using highly selective classifiers on the *bottom-left corner* of the ROC curve that accept few outputs—i.e., F $\approx$ 0. For intuition, consider an ROC that is linear around the origin, i.e. T(F) = $\alpha$F for F $\ll$ 1. In that case, Proposition 1 implies that the derivative of the performance $A(C)$ with respect to expected compute $C$ is zero for small F – meaning that performance eventually plateaus as compute increases. More generally, our next proposition shows that the large-scale performance of RS is determined by the derivative of the ROC curve at the origin whenever T(0) = 0.

**Proposition 2.** *Let $f$ be a verifier and $T : [0,1] \mapsto [0,1]$ be the ROC curve of $f$. If $T(0) = 0$ and $T(F)$ is continuously differentiable in a neighborhood of $F = 0$, we have*

$$\lim_{C \to \infty} A(C) = \frac{T'(0) \cdot \pi}{T'(0) \cdot \pi + (1 - \pi)}. \tag{8}$$

*Otherwise if $T(0) > 0$, $C(0)$ is finite and $\bar{A}(0) = A(C(0)) = \lim_{C \to C(0)} A(C) = 1$.*

The proof of Proposition 2 follows directly from Equation (6) using Taylor's theorem and L'Hospital (see Appendix C.3). Proposition 2 is in line with the large-scale behavior shown in Figure 2a: As both ROC curves have the same slope at the origin, their large-scale performance is the same. The opposite is observed in Figure 1: Here, the ROC curves have different slopes near the origin, but are similar in the top-right corner. As predicted by Proposition 2, RS performs substantially worse in the high-compute limit when the ROC curves have smaller slope near the origin.

### 4.3 CAN RS PERFORMANCE BE EXTRAPOLATED?

So far we established in Equation (7) and Proposition 2 that the scaling in the low- and high-compute setting is determined by the local geometry of the ROC curve in the top-right vs the bottom-left corner respectively. As the geometry in neither of the corners puts strong constraints on the geometry in the other corner, without *a priori* knowledge of the ROC curve's behavior near the origin, we cannot predict large-scale performance based on small-scale performance. Our next proposition formalizes this intuition.

**Proposition 3.** *Fix a compute budget $B < \sup_F C(F) =: C_{max}$ with $A_f(B) > 0$, and suppose the RS accuracy $A_f(C)$ is known for all $F$ with $C(F) \leq B$ for a fixed, unknown verifier $f$. Then:*

1. *There exist verifiers $f_0, f_1$ consistent with the accuracies observed up to compute $B$, such that $\lim_{C \to C_{max}} A_{f_0}(C) = 0$ and $\lim_{C \to C_{max}} A_{f_1}(C) = 1$ respectively.*

2. *Assuming $f$ has a concave ROC curve and the left one-sided derivative of $A(C)$ is strictly positive at $C = B$, there are consistent verifiers $\tilde{f}_{A(B)}, \tilde{f}_1$ with concave ROC curves such that $\lim_{C \to C_{max}} A_{\tilde{f}_{A(B)}}(C) = A(B)$ and $\lim_{C \to C_{max}} A_{\tilde{f}_1}(C) = 1$ respectively.*

We prove Proposition 3 in Appendix C.5. It implies that observing RS performance at small scales, we can usually not distinguish between the two cases of i) no further performance gains from scaling and ii) eventual perfect performance. This can be observed empirically in Figure 1, where both verifiers lead to the same performance at small compute, but performance diverges at high-compute. Figure 2b shows an even more extreme case: While RS initially scales substantially faster for the blue ROC curve, performance quickly stagnates. Correspondingly, RS with the orange ROC curve reaches substantially higher performance levels, despite the slower initial scaling.

## 5 BEST-OF-N

While the scaling of RS has a clear and simple dependence on the ROC curve, the amount of compute used by RS is random, and its expectation only implicitly depends on the chosen threshold $\tau$. Compared to that, BoN gives users the ability to explicitly specify a deterministic amount of compute $N$. In this section, we switch our focus to the scaling behavior of Best-of-N (BoN).

As in the previous section, we treat the verifier $f$ as fixed and write $g_N$ rather than $g_N^f$ to denote the output distribution of BoN as defined in (3). The accuracy of BoN is equal to the probability that the highest-scoring sample $X^\star$ among $N$ draws is a correct answer, i.e. $\text{ACC}(g_N) = P_{g_N}[y(X^\star) = 1]$. To characterize BoN performance, we make following regularity assumption about the score $f(x)$:

**Assumption 1.** *The distribution of scores $f(X)$ is either discrete or absolutely continuous with respect to the Lebesgue measure (i.e. has a density).*

This allows us to show that the ROC curve of the verifier $f$ again determines how the accuracy of BoN scales with compute. While RS is governed by the *local* geometry of the ROC curve, we will

see that the scaling behavior of BoN depends on the *global* properties. We begin by defining the probability that BoN produces a correct answer, conditional on $p$ of the $N$ samples being correct,

$$H(k, p) := P_{g_{\text{base}}}\left[ y\left( \underset{\{X_i\}_{i \in [k+p]}}{\arg\max} f(X_i) \right) = 1 \,\middle|\, \sum_{i \in [N]} y(X_i) = p \right].$$

It then follows that the accuracy of BoN can be expressed by

$$\text{ACC}(g_N) = \mathbb{E}_{p \sim B(\pi, N)}[H(N - p, p)], \tag{9}$$

where $B(\pi, N)$ denotes the binomial distribution with success probability $\pi$ and $N$ trials.

We note that $H(1, 1)$ equals to the area under the ROC curve (AUROC). Inspired by Scherlis (2021), we find that $H(k, p)$ can be written as a weighted integral over the ROC curve for any $k$ and $p$. This allows us to cast the BoN accuracy $\text{ACC}(g_N)$ as an integral over the ROC curve:

**Proposition 4.** *Let $f$ be a score with ROC curve $T(F)$ for which Assumption 1 holds. Define $\psi(F) := (1 - \pi)(1 - F) + \pi(1 - T(F))$. Then for $N \geq 2$, the accuracy of BoN is given by*

$$ACC(g_N) = 1 - (1 - \pi)N \int_0^1 \psi(F)^{N-1} dF. \tag{10}$$

*If $T(F)$ is concave, the BoN accuracy $ACC(g_N)$ is non-decreasing in $N$.*

We prove Proposition 4 in Appendix C.6. Note that in the case of a perfect verifier $y = f$, we have $T(F) = 1$ and thus $\psi(F) = (1 - \pi)(1 - F)$, such that Proposition 4 yields the well-known formula $1 - (1 - \pi)^N$ for pass@N. In addition, the proposition implies that when the ground truth $y(x)$ is binary, overoptimization (Gao et al., 2023)—where actual performance worsens as more samples are used— can only be a problem for BoN when the verifier's ROC curve is non-concave.

### 5.1 LOW-COMPUTE REGIME

For $N = 2$, noting that $H(0, 2) = 1$, $H(2, 0) = 0$, the expectation in Equation (9) is fully determined by the area under the ROC curve $H(1, 1)$ and the original task performance $\pi$. In this case, we obtain a simple formula for the performance gain going from Best-of-1 to Best-of-2 sampling:

$$\text{ACC}(g_2) - \text{ACC}(g_1) = \pi^2 + 2\pi(1 - \pi)H(1, 1) - \pi$$

For a random-equivalent verifier, $H(1, 1) = 0.5$ such that the performance gain equals zero, while for the perfect verifier $f = y$ the gain equals $\pi(1 - \pi)$. Notably, this maximal possible "slope" of $\pi(1 - \pi)$ is substantially below the same slope of $\pi$ for RS at $C = F = T = 1$, suggesting that RS might outperform BoN when controlling for compute. Our next proposition confirms this:

**Proposition 5.** *Let $f$ be a verifier with concave ROC curve $T(F)$ for which Assumption 1 holds. Set $F_N$ to be the solution to $C(F_N) = N$ and fix any $N \in \mathbb{N}$ for which $F_N$ exists. Then, RS with the verifier $h_{F_N}$ outperforms BoN, i.e. $ACC(g^{h_{F_N}}) \geq ACC(g_N)$.*

We prove Proposition 5 in Appendix C.9. Interestingly, the advantage of RS vanishes in the limit: In the next subsection, we analyze the large-scale limit of BoN, and show that it matches the performance of RS we established in Proposition 2.

### 5.2 HIGH-COMPUTE REGIME

While the integral formula for the performance of BoN from Proposition 4 is harder to analyze than the more *local* formula for the performance of RS from Equation (6), our next theorem shows that both methods still perform the same in the large scale limit.

**Theorem 1.** *In the setting of Proposition 4, assume that $T(F)$ is continuously differentiable in a neighborhood of $F = 0$. Then if $T(0) > 0$, we have $\lim_{N \to \infty} ACC(g_N) = 1$. Otherwise if $T(0) = 0$,*

$$\lim_{N \to \infty} ACC(g_N) = \frac{T'(0) \cdot \pi}{T'(0) \cdot \pi + (1 - \pi)}.$$

Theorem 1 is proven in Appendix C.7. Intuitively, for large $N$, $\text{ACC}(g_N)$ is mostly determined by the largest values of $\psi(F) = (1 - \pi)(1 - F) + \pi(1 - T(F))$, which correspond to small $F$ and $T(F)$. Thus, the limiting behavior of BoN is determined by behavior of $T(F)$ near the origin.

Comparing with Proposition 2, the high-compute limit of BoN performance $\text{ACC}(g_N)$ is equal to the high-compute limit for RS. Combined with Proposition 8, this suggests that Theorem 1 might point to a more fundamental limit to the performance gains achievable with imperfect verifiers. As RL algorithms are often trained to mimic the resampling distribution (Xiong et al., 2025), such limit might also extend to RL. In Appendix B.3, we further explore this connection, casting the RS distribution as the zero-penalty limit of KL-constrained RL.

### 5.3 CAN BoN PERFORMANCE BE EXTRAPOLATED?

As in Section 4.3 we investigate whether it is possible to extrapolate BoN performance from observations at low compute without knowing the ROC curve. We again provide a negative result: Despite its smoother scaling, the limiting performance of BoN remains impossible to predict from small-scale observations, especially when the ROC curve cannot be guaranteed to be concave.

**Proposition 6.** *Consider a verifier $f$ satisfying Assumption 1 such that $\lim_{N \to \infty} ACC(g_N^f) = c < 1$. Then for any $n \in \mathbb{N}$ and $\epsilon > 0$, there are verifiers $\tilde{f}_0, \tilde{f}_1$ satisfying Assumption 1 such that $|ACC(g_N^f) - ACC(g_N^{\tilde{f}_i})| \le \epsilon$ for all $N \le n$ and $i \in \{0, 1\}$, but*

$$\lim_{N \to \infty} ACC(g_N^{\tilde{f}_0}) = 0 \quad while \quad \lim_{N \to \infty} ACC(g_N^{\tilde{f}_1}) = 1.$$

*If $f$ has a concave ROC curve, the verifier $\tilde{f}_1$ can be chosen to have a concave ROC curve as well.*

We prove Proposition 6 in Appendix C.12. Analogously to Proposition 3, it shows that without further assumptions, any early scaling in BoN is compatible with both zero and perfect performance $\text{ACC}(g_N)$ in the large $N \to \infty$ limit. Even assuming concavity, it remains impossible to derive meaningful upper bounds on large scale performance by extrapolating from smaller scales.

## 6 EXPERIMENTS

In this section, we evaluate a series of open-weight instruction-tuned LLMs from the `Qwen3` (Yang et al., 2025) and `LLama` (Grattafiori et al., 2024) families as both generators and verifiers on questions from GSM8K (Cobbe et al., 2021) and MATH500 (Hendrycks et al., 2021). To generate an answer $x$, we use few-shot prompting with 5 random train examples and temperature $t = 1$. For verification, we prompt models to score answer correctness from 0 to 10, after employing a chain of thought (Tian et al., 2023; Cruz et al., 2024) and normalize the scores to lie in $[0, 1]$. To increase the resolution of the score, we repeat this process five times per answer and use the average of the responses as the final score $f(X)$. Further implementation details are described in Appendix E.

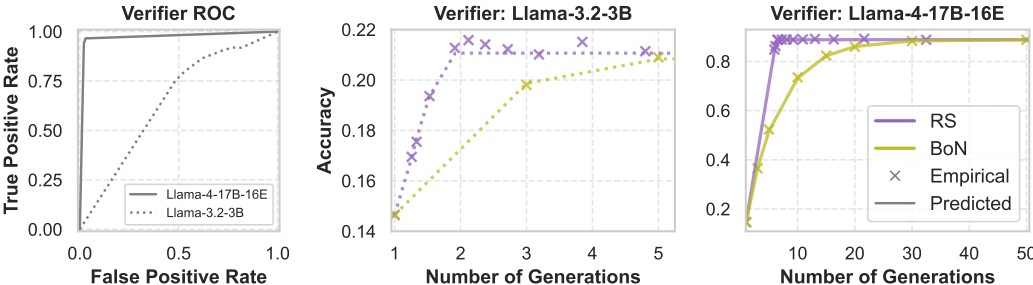

Figure 3: Empirical performance (× markers) of RS (purple) and BoN (olive) on GSM8K test question 2, overlaid with theoretical predictions (lines). Dotted: `Llama-3.2-3B` as verifier (single COT). Solid: `Llama-4-17B-16E` as verifier (single COT). Controlling for the number of generated samples, RS consistently outperforms BoN for both verifiers. Generator: `Llama-3.2-3B`.

In Sections 4 and 5, we rigorously characterized how the *per-instance* generative accuracy of BoN and RS scales with test-time compute. Correspondingly, we now validate the theoretical predictions

for BoN and RS on individual queries. For a small set of queries, we sample and score 1000 responses using different `Qwen3` and `LLama` models as both generators and verifiers. These samples are used to both i) simulate BoN and RS by resampling and ii) estimate different verifiers' ROC curves. We then use these ROC estimates to predict the accuracies for both RS and BoN using Equation (5) and Equation (10) respectively. Note, that RS often terminates after a small number of samples, even at the maximal threshold of $\tau = 1$. To simulate RS for larger numbers of samples, we thus randomly interpolate between the classifier $h^1$ and the (always rejecting) classifier $h^2$. Due to Proposition 1, RS with the resulting classifier uses more samples than $h^1$, but is no more accurate.

Across the board, our theory predicts both methods' accuracies with high precision (see results on additional questions in Appendix E). Exemplarily, Figure 1 shows the results for a `Qwen3 1.7B` generator on GSM8K question $i = 58$, which illustrates the issues discussed in Proposition 3 particularly well. As predicted by Proposition 1, since the ROC curve plateaus near $F = 1$ (top-right portion), early RS scaling follows the same linear trend for all verifiers. Similarly BoN performance is almost identical up until $N = 3$, but diverges at higher compute. This is in line with Proposition 2, predicting that different ROC slopes at $F = 0$ lead to different performance levels at high compute. Comparing the middle and right panel, we can also see that RS outperforms BoN for fixed compute.

This can be observed more directly in Figure 3, which shows results for a `Llama-3.2-3B` generator on GSM8K question $i = 2$ with verifiers using a single chain of thought. We plot RS and BoN performance for the same verifier in the same panel and observe a stark difference between both methods' scaling, as predicted by Proposition 5. Notably, RS consistently outperforms BoN, despite some non-concavity in the ROC curve of `Llama-3.2-3B`. While the relative prediction error becomes noticeable in the middle panel, the absolute errors remain below one percentage point.

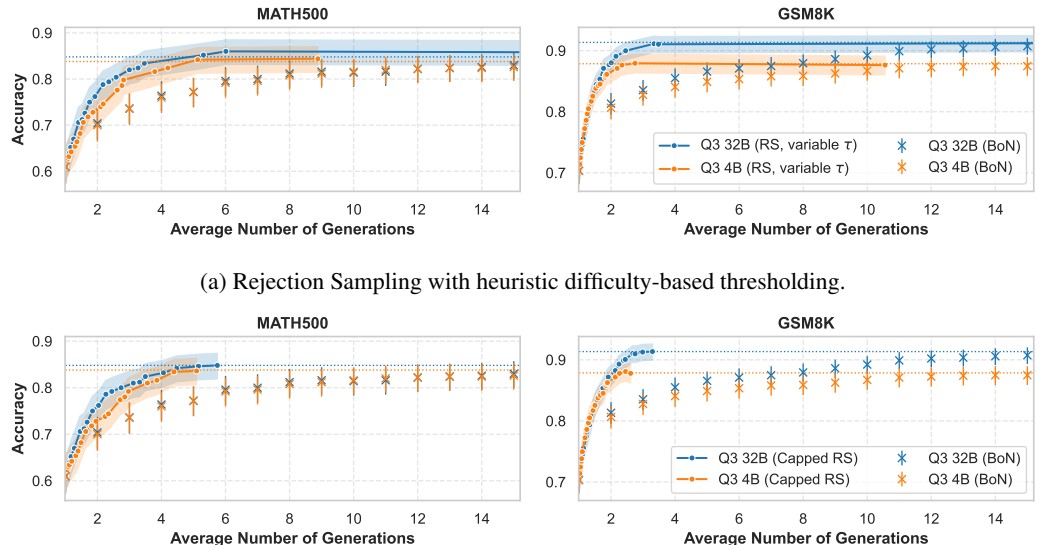

(a) Rejection Sampling with heuristic difficulty-based thresholding.

(b) Capped Rejection Sampling.

Figure 4: Aggregate accuracy of BoN vs RS on MATH500 (left) and GSM8K (right). Dotted lines: Bo25 performance. Controlled for compute RS consistently outperforms BoN. Verifiers: `Qwen3-32B` (blue), and 4B (orange). Generator: `Qwen3-1.7B`. Error bars: Exact 90% CIs.

**Aggregate results** Our theoretical results on the predictability of scaling and the dominance of RS over BoN are restricted to fixed queries. However, one might intuitively expect them to also apply on aggregate over a larger dataset $\mathcal{D}$ of queries. To test this, we run RS and BoN on each query $q$ in the GSM8K and MATH500 test sets for given thresholds $\tau(q)$ and numbers of BoN samples $N$. As mentioned before, choosing a good threshold $\tau(q)$ for a given query $q$ can be challenging. However, a separate problem arises when using RS over multiple queries: Using the same threshold threshold $\tau$ for all queries can cause large variance in the false positive rates $F$ and compute costs $C(F)$ across queries. In practice, some queries $q$ receive high scores $f(x)$ for most answers $x$ and thus require large thresholds $\tau \approx 1$ to reliably identify correct answers. Meanwhile as shown in Figure E.1,

scores $f(x) \approx 1$ are exceedingly rare for a small set of hard queries $q$, such that running RS with a high constant threshold for the whole dataset may run indefinitely. To address this, we consider two approaches: a difficulty-based heuristic to adjust the threshold $\tau$ per query, and a "hybrid" approach that starts with RS for a constant $\tau$, but reverts to BoN after a fixed number of samples.

First, we implemented a simple two-tier heuristic based on query difficulty: For $t \in [0, 1]$, we use the threshold $\tau(q) = t$ for "easy" queries where the generator $g$ produces correct answers $x$ at least $1\%$ of the time, and the threshold $\tau(q) = \frac{t}{2}$ for the remaining "hard" queries. Figure 4a plots this method and BoN's average accuracy against their respective average number of generated samples, with error bars indicating $90\%$ confidence intervals for accuracy. The figure replicates several of our key observations from the single-query setting: On both datasets, controlling for compute, RS outperforms BoN, but the gap between the methods vanishes at larger compute levels. Additionally on GSM8K, performance for the Qwen3-4B and Qwen3-32B verifiers is the same at low compute, but a noticeable gap between the verifiers emerges at larger compute. This indicates that we cannot rely on extrapolation to predict performance for high levels of test-time compute.

However, difficulty-based thresholds $\tau(q)$ are challenging to implement in practice, when query difficulty is unknown. A promising alternative are "hybrid" methods that use a threshold $\tau$ that works well for most queries, while limiting the compute wasted on queries for which that threshold is rarely reached. A simple approach is "Capped Rejection Sampling": We fix the same threshold $\tau$ for every query and sample answers $x$ until either $f(x) \geq \tau$, or we have sampled 25 times. In the latter case, we return the answer $x$ with the highest score $f(x)$ among our samples. Figure 4b shows this works well in practice: In all cases, CRS matches Bo25 performance, but requires substantially fewer samples. Furthermore, Capped RS performs very similar to the variable threshold heuristic proposed in the previous paragraph, while arguably being simpler to implement. Additional experiments on GPQA (STEM reasoning) and HumanEval+ (Code generation) can be found in Appendix E.

## 7 DISCUSSION

Our results precisely quantify the limitations of resampling with imperfect verifiers. Both theoretical and empirical results indicate limited dependence between verifier performance at low and high test-time compute, cautioning against trend extrapolation as a means to predict performance. Our work opens up several new lines of inquiry: On the theoretical side, future work could explore how distributional assumptions — on the per-item accuracies $\pi$ of the generator $g_{\text{base}}$ and the per-item ROC curves — affect our conclusions regarding extrapolability and the dominance of RS over BoN. On the methodological side, further refinement and analysis of "hybrid" methods is a promising avenue. In addition, the dependency of early and later test-time-scaling on different regions of the ROC curve motivates future work on training customized verifiers for different test-time budgets.

| Symbol | Description |
|---|---|
| $q$ | A query or prompt given to the generator |
| $g$ | A generator model (e.g., an LLM) |
| $P_g$ | The distribution induced by a generator $g$ |
| $x$ | A text response sampled from $g$, i.e., $x \sim P_g$ |
| $f : \mathcal{X} \to [0, 1]$ | Score function estimating the correctness of $x$ |
| $h^\tau(x)$ | Classifier that accepts if $f(x) \geq \tau$ |
| $\mathcal{H}(f)$ | The set of all classifier $h^\tau$ induced by thresholding the verifier $f$ |
| $g_N^f$ | Best-of-N sampler (for score $f$) |
| $g_N$ | Best-of-N sampler (when score $f$ is clear from context) |
| $g^h$ | Rejection-sampled generator: sample $x \sim P_g$ until $h(x) = 1$ |
| $y : \mathcal{X} \to \{0, 1\}$ | Ground-truth label indicating whether $x$ is a correct response |
| $\text{ACC}(g)$ | Accuracy of a generator: $\Pr_{x \sim P_g}[y(x) = 1]$ |
| $\pi$ | Accuracy $\text{ACC}(g_{\text{base}})$ for the base generator $g_{\text{base}}$ |
| $\text{T}(g, h)$ | True positive rate of classifier $h$ under $P_g$ |
| $\text{F}(g, h)$ | False positive rate of classifier $h$ under $P_g$ |
| $\text{T}(\cdot) : [0, 1] \to [0, 1]$ | ROC curve given by $\text{T}(F) = \max\{\text{T}(g_{\text{base}}, h) : h \in \mathcal{H}(f), \text{F}(g_{\text{base}}, h) = F\}$. |
| $h_\text{F}$ | Classifier $h$ with best T, given F: $\arg\max_{h \in \mathcal{H}(f): \text{F}(g,h)=\text{F}} \text{T}(g, h)$ |
| $N(\text{F})$ | Number of samples drawn from $g$ until first accepted by $h_\text{F}$ |
| $C(\text{F})$ | Expected number of samples before acceptance: $\mathbb{E}(N(\text{F}))$ |
| $\bar{A}(\text{F})$ | Accuracy $\text{ACC}(g^{h_\text{F}})$ viewed as a function of $\text{F}(g, h_\text{F}) = \text{F}$ |
| $A(C)$ | Accuracy $\bar{A}(\text{F})$ viewed as a function of $C(\text{F})$ |
| $\psi(\text{F})$ | Rejection probability $(1 - \pi)(1 - F) + \pi(1 - \text{T}_f(F))$ |

Table 1: Primary Notation

## 8 REPRODUCIBILITY STATEMENT

We clearly document our experimental setup in Section 6 and Appendix E. All code necessary to reproduce experiments is available at the following repository: `https://github.com/socialfoundations/roc-n-reroll`.

## 9 LLM USAGE

We made use of LLMs for general research support. In particular, we used them as a tool to find additional relevant literature for related work, and to help with brainstorming proof ideas. Most relevant to this, GPT-o3 suggested the use of the layer-cake trick to deal with some of the integrals appearing in the BoN analysis, allowing us to substantially simplify some of our proofs. All text, including proofs, was written by the authors.

## ACKNOWLEDGMENTS

Florian Dorner is grateful for financial support from the Max Planck ETH Center for Learning Systems (CLS). The authors thank the International Max Planck Research School for Intelligent Systems (IMPRS-IS) for supporting André F. Cruz.

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

# A ADDITIONAL RELATED WORK

## A.1 "REASONING" METHODS

Reasoning models that are post-trained via RL and generally use more test-time compute play a large role in industry (OpenAI, 2024; Kimi et al., 2025; Guo et al., 2025; Kavukcuoglu, 2025). While academic efforts to reproduce RL training at smaller scales exist (Zeng et al., 2025), some of the more successful reasoning models from academia are based on model distillation (Muennighoff et al., 2025; Team, 2025). These models are trained via supervised fine-tuning on outputs generated by larger reasoning models with the goal of learning to copy the larger models' behavior. Reminiscent of earlier chain-of-thought prompting (Wei et al., 2022; Kojima et al., 2022) designed to make models "think step-by-step", Muennighoff et al. (2025) show that the performance of distilled models can sometimes be boosted by simple modifications to the generation process: Forcing the model to generate longer answers by repeatedly replacing the "end-of-thinking" token with the word "Wait" noticeably improved their model's performance on the AIME 2024 benchmark. In contrast, Wu et al. (2025) provide empirical evidence that too large answer length can be detrimental, as well as a simple theoretical model to explain this phenomenon.

However, it is not clear whether reasoning methods provide a fundamental improvement compared to resampling-based scaling methods, or merely allow for inference compute to be partially *amortized*: Yue et al. (2025) show that while reasoning models initially outperform base models, this trend reverses when resampling methods with perfect verifiers are applied to both models at large compute budgets.

## A.2 ROC CURVES

Classification algorithms usually operate by learning a score $f(x)$ that induces a set of classifiers based on applying different decision thresholds to the score. For a given score, the ROC curve represents possible tradeoffs between the induced classifiers' false and true positive rate (consider Fawcett (2006) for a summary of key properties). The area under the ROC curve (AUROC) is a common metric for classifier performance, and equals the probability of giving a higher score to a randomly selected positive instance than to a randomly selected negative one. Davis and Goadrich (2006) note that there is a bijection between ROC curves and precision-recall tradeoffs. While precision is equivalent to the accuracy of rejection sampling in our setting, we focus on the tradeoff between precision and the expected number of samples for rejection sampling, rather than recall.

Noting that certain score ranges usually do not have any clinical meaning, Dodd and Pepe (2003) suggest to consider the *partial area under the ROC curve*, focusing on a subinterval of false positive rates. More recently Shi et al. (2024) propose the *lower-left partial area under the ROC curve* that additionally caps the true positive rate, and show that this metric can be used to provide bounds on top-k ranking metrics. In contrast, our results establish that the limiting accuracy of both rejection sampling and BoN are fully determined by the *slope* of the ROC curve, close to the origin.

# B ADDITIONAL RESULTS

## B.1 TOKEN-BASED COMPUTE VS COMPLETION-BASED COMPUTE

Let $T_i$ denote the iid number of tokens generated by both the generator and verifier for the $i$-th sample, and let $K$ be the total number of samples drawn by RS. Since $T_j$ is independent of the event $\{K = i\}$ for $j \geq i$, Wald's theorem (Mitzenmacher and Upfal, 2017, p. 300-301) implies that:

$$\mathbb{E}\left[\sum_{i=1}^{K} T_i\right] = \mathbb{E}[K] \cdot \mathbb{E}[T]$$

Thus, when compute is measured in terms of expected number of generated tokens rather than samples, RS's compute is simply multiplied by $\mathbb{E}[T]$. The same is straightforwardly true for BoN.

## B.2 A CONCISE FORMULA FOR BON SCALING

We show that for concave ROC curves, we can substantially simplify the integral representation for $ACC(g_N)$ from Proposition 4.

**Proposition 7.** *Let $f$ be a score with concave ROC curve $T(F)$ for which assumption 1 holds and denote $\psi(F) = (1 - \pi)(1 - F) + \pi(1 - T(F))$. Then the best-of-N accuracy can be written as*

$$ACC(g_N) = 1 - (1 - \pi)(N^2 - N) \int_0^{1 - \pi T(0)} \psi^{-1}(a) a^{N-2} da. \tag{11}$$

We prove Proposition 7 in Appendix C.8. We note that this is closely related to the expectation of $\psi^{-1}(a)$ for $a \sim \beta(N - 1, 1)$, as the beta distribution $\beta(N - 1, 1)$ has the density $(N - 1)a^{N-2}$.

Next, we use that according to Proposition 7 $ACC(g_N)$ is linear in the inverse $\psi^{-1}$ of the reweighed ROC curve $\psi(F)$. This allows us to focus on computing the BoN accuracy for a "basis" of all possible $\psi^{-1}$ and then compute $ACC(g_N)$ as a linear combination of the "basis" functions' accuracies. In particular, for piece-wise linear as well as concave ROC curves, we obtain a simple representation in the following theorem:

**Theorem 2.** *Let $f$ be a verifier with piece-wise linear concave ROC curve $T(F)$ for which assumption 1 holds. Then there are positive weights $w_i$ with $\sum_{i=0}^{\infty} w_i = 1$, $x_i \leq 1$ and $\sum_{i=0}^{\infty} w_i \frac{1}{x_i} \leq \frac{1}{1-\pi}$ such that for all $N$*

$$ACC(g_N) = 1 - (1 - \pi) \sum_{i=0}^{\infty} w_i x_i^{N-1}. \tag{12}$$

*Conversely, for any $w_i$ and $x_i$ respecting the constraints above, there exists a verifier $f$ with concave ROC curve $T(F)$ such that Equation (12) holds for all $N$.*

We prove Theorem 2 in Appendix C.10. From Equation (12), it is easy to see that $ACC(g_N)$ is non-decreasing for piece-wise linear concave ROC curves. Proposition 9 is then proven by approximating any concave ROC curve with piece-wise linear ones.

## B.3 RS AND REINFORCEMENT LEARNING

Recently, Xiong et al. (2025) have demonstrated that fine-tuning an LLM to match its own rejection sampling distribution for a given reward $r$ is competitive with reinforcement learning (RL) using the same reward. Our next proposition provides a theoretical justification for this observation, showing that the optimal KL-regularized RL policy converges to RS, as we let regularization go to zero:

**Proposition 8.** *For answers $x$ in a discrete space $\mathcal{X}$, let $g^*(\beta)$ be the optimal solution to the KL-regularized reinforcement learning problem $\arg\max_{g^*(\beta)}[\mathbb{E}_{g^*(\beta)}[r(X)] - \beta\mathbb{D}_{KL}[P_{g^*(\beta)}||P_{g_{base}}]]$ for $\beta > 0$ and $r(x) = \mathbb{I}[f(x) \geq \tau]$, where $\mathbb{I}$ is the indicator function. Then for any $x \in \mathcal{X}$,*

$$\lim_{\beta \to 0} P_{g^*(\beta)}[X = x] = P_{g_{base}}[X = x | f(x) \geq \tau].$$

We prove Proposition 8 in Appendix C.4. Combined with Xiong et al. (2025), it suggests that the limiting performance of generators trained by RL might be given by Equation (8), just as for RS.

## C PROOFS

### C.1 PROOF OF PROPOSITION 1

*Proof.* Since $T(F)$ is increasing, $C(F) = \frac{1}{T(F) \cdot \pi + F \cdot (1 - \pi)}$ is a strictly decreasing function and thus invertible function of F. Correspondingly, we can write $F(z)$ as a well-defined function of $z = C(F)$. In particular, this means that the performance $A(F)$ can be written as $A(F(z))$, i.e. a function of the runtime $z = C(F)$.

| Symbol | Description |
|---|---|
| $\psi(F)$ | Rejection probability $(1-\pi)(1-F)+\pi(1-\mathrm{T}_f(F))$ |
| $\psi^{-1}(a)$ | Inverse of rejection probability $\psi(a)$ |
| $\hat{\psi}^{-1}(a)$ | Domain-extended version of $\psi^{-1}$ (zero when $\psi^{-1}(a)$ is not defined) |
| $\hat{b}_x^{-1}(a)$ | Hinge function at $x$ (used as a "basis" to approximate convex $\hat{\psi}^{-1}$) |
| $\mathrm{ACC}(g_N^{\hat{\psi}^{-1}})$ | Bon accuracy $\mathrm{ACC}(g_N^f)$ parameterized by $\psi^{-1}$ |

Table 2: Additional Notation

We compute the derivative

$$
\begin{aligned}
\frac{dA(C)}{dC}\bigg|_{C=C(\mathrm{F})} &= \frac{d\bar{A}(\mathrm{F}(C))}{dC}\bigg|_{C=C(\mathrm{F})} \\
&= \frac{d\bar{A}(x)}{dx}\bigg|_{x=\mathrm{F}} \frac{d\mathrm{F}(C)}{dC}\bigg|_{C=C(\mathrm{F})} \\
&= \frac{d\bar{A}(x)}{dx}\bigg|_{x=\mathrm{F}} \frac{d\mathrm{F}(C)}{dC}\bigg|_{C=C(\mathrm{F})} \\
&= \frac{d\bar{A}(x)}{dx} \frac{1}{\frac{dC(x)}{dx}}\bigg|_{x=\mathrm{F}}.
\end{aligned}
$$

We calculate

$$
\frac{dC(x)}{dx} = \frac{\pi - 1 - \pi \mathrm{T}'(x)}{(x(\pi-1)-\pi \mathrm{T}(x))^2}
$$

and

$$
\frac{d\bar{A}(x)}{dx} = \frac{\pi\left((\pi \mathrm{T}(x)-x(\pi-1))\mathrm{T}'(x)-(\pi \mathrm{T}'(x)-\pi+1)\mathrm{T}(x)\right)}{(x(\pi-1)-\pi \mathrm{T}(x))^2}
$$

plug them back into the original derivative, we have

$$
\begin{aligned}
\frac{dA(C)}{dC}\bigg|_{C=C(\mathrm{F})} &= \frac{\pi\left((\pi \mathrm{T}(\mathrm{F})-\mathrm{F}(\pi-1))\mathrm{T}'(\mathrm{F})-(\pi \mathrm{T}'(\mathrm{F})-\pi+1)\mathrm{T}(\mathrm{F})\right)}{\pi-1-\pi \mathrm{T}'(\mathrm{F})} \\
&= \pi \frac{(\mathrm{F}(1-\pi))\mathrm{T}'(\mathrm{F})-(1-\pi)\mathrm{T}(\mathrm{F})}{\pi-1-\pi \mathrm{T}'(\mathrm{F})} \\
&= \pi \frac{(1-\pi)\mathrm{T}(\mathrm{F})-(\mathrm{F}(1-\pi))\mathrm{T}'(\mathrm{F})}{1+\pi \mathrm{T}'(\mathrm{F})-\pi} \\
&= \pi \frac{(1-\pi)(\mathrm{T}(\mathrm{F})-\mathrm{FT}'(\mathrm{F}))}{1+\pi \mathrm{T}'(\mathrm{F})-\pi}.
\end{aligned}
$$

This is positive, if and only if $\mathrm{T}(\mathrm{F})-\mathrm{FT}'(\mathrm{F}) \geq 0$. Assuming the ROC curve is concave, it has to be continuous, while $\mathrm{T}'(\mathrm{F})$ is non-increasing.

Thus:

$$
\mathrm{T}(\mathrm{F}) = \int_0^{\mathrm{F}} \mathrm{T}'(t)dt \geq \mathrm{F} \cdot \mathrm{T}'(\mathrm{F}),
$$

which implies:

$$
\mathrm{T}(\mathrm{F})-\mathrm{FT}'(\mathrm{F}) \geq 0,
$$

with both inequalities strict for $\mathrm{F} > 0$ if the ROC curve is strictly concave.

$\square$

## C.2 PROOF OF EQUATION (7)

*Proof.* We plug in $T = F = 1$ into the expression of the derivative of the performance–compute scaling curve and get:

$$
\begin{aligned}
\left.\frac{dA(C)}{dC}\right|_{C=1} &= \pi \left.\frac{(F(1-\pi))\,T'(F) - (1-\pi)\,T(F)}{\pi - 1 - \pi T'(F)}\right|_{F=T(F)=1} \\
&= \pi \frac{((1-\pi))\,T'(1) - (1-\pi)}{\pi - 1 - \pi T'(1)} \\
&= \pi \frac{\pi - 1 - \pi T'(1) + T'(1)}{\pi - 1 - \pi T'(1)}.
\end{aligned}
$$

□

## C.3 PROOF OF PROPOSITION 2

*Proof.* We first focus on the case of $T(0) = 0$. Using Taylor's Theorem, we get

$$
T(F) = F\frac{dT}{dF}|_{F=0} + o(F) = T'(0)F + o(F),
$$

where the derivative exists and is continuous in a neighborhood of $F = 0$, by assumption. Now,

$$
\begin{aligned}
\lim_{C\to\infty} A(C) &= \lim_{F\to 0} \bar{A}(F) \\
&= \lim_{F\to 0} \frac{\pi \cdot T(F)}{\pi \cdot T(F) + (1-\pi)\cdot F} \\
&= \lim_{F\to 0} \frac{\pi \cdot T'(0)F}{\pi \cdot T'(0)F + o(F) + (1-\pi)\cdot F} + \frac{\pi o(F)}{\pi \cdot T'(0)F + o(F) + (1-\pi)\cdot F} \\
&= \lim_{F\to 0} \frac{\pi \cdot T'(0)F}{\pi \cdot T'(0)F + o(F) + (1-\pi)\cdot F} \\
&= \lim_{F\to 0} \frac{\pi \cdot T'(0)}{\pi \cdot T'(0) + (1-\pi)}
\end{aligned}
$$

When $T(0) > 0$, we simply get

$$
\lim_{C\to C(0)} A(C) = \bar{A}(0) = \frac{T(0)\mathrm{ACC}(g^h)}{T(0)\mathrm{ACC}(g^h)} = 1.
$$

□

## C.4 PROOF OF PROPOSITION 8

*Proof.* Due to Rafailov et al. (2023), $g^*(\beta)$ has to fulfill

$$
P_{g^*(\beta)}[X = x] = \frac{1}{Z}P_{g_{\mathrm{base}}}[X = x]\exp[\frac{1}{\beta}r(X = x)],
$$

where $Z$ is a normalization constant. We can decompose $Z$ as follows:

$$
\begin{aligned}
Z &= \sum_x P_{g_{\mathrm{base}}}[X = x]\exp[\frac{1}{\beta}r(x)] \\
&= \sum_{x:r(x)=1} P_{g_{\mathrm{base}}}[X = x]\exp(\frac{1}{\beta}) + \sum_{x:r(x)=0} P_{g_{\mathrm{base}}}[X = x]\exp(0) \\
&= \exp(\frac{1}{\beta}) \sum_{x:r(x)=1} P_{g_{\mathrm{base}}}[X = x] + \sum_{x:r(x)=0} P_{g_{\mathrm{base}}}[X = x] \\
&= \exp(\frac{1}{\beta})P_{g_{\mathrm{base}}}[r(X) = 1] + P_{g_{\mathrm{base}}}[r(X) = 0].
\end{aligned}
$$

Now for any $x$ with $r(x) = 0$, we have

$$P_{g^*(\beta)}[X = x] = \frac{P_{g_{\text{base}}}[X = x]}{\exp(\frac{1}{\beta})P_{g_{\text{base}}}[r(X) = 1] + P_{g_{\text{base}}}[r(X) = 0]},$$

which goes to zero for $\beta \to 0$. Similarly, for $x$ with $r(x) = 1$, we have

$$P_{g^*(\beta)}[X = x] = \frac{\exp(\frac{1}{\beta})P_{g_{\text{base}}}[X = x]}{\exp(\frac{1}{\beta})P_{g_{\text{base}}}[r(X) = 1] + P_{g_{\text{base}}}[r(X) = 0]},$$

which converges to

$$\frac{P_{g_{\text{base}}}[X = x]}{P_{g_{\text{base}}}[r(X) = 1]} = P_{g_{\text{base}}}[X = x | r(X) = 1]$$

as $\beta \to 0$. Thus, as the KL penalty $\beta$ goes to zero, the probability mass function of the optimal RL policies $g^*(\beta)$ converge to the probability mass of the conditional distribution $P_{g_{\text{base}}}[x | r(x) = 1]$ in a point-wise sense. □

## C.5 PROOF OF PROPOSITION 3

We first prove a useful lemma that casts the accuracy of rejection sampling as a function of the ratio $\alpha = \frac{T}{F}$:

**Lemma 1.** *Whenever $T(F) = \alpha F$, the accuracy of the corresponding rejection-sampled distribution $g^h$ is:*

$$\bar{A}(F) = \frac{\alpha \cdot \pi}{\alpha \cdot \pi + (1 - \pi)}$$

*Proof.* Plug in $T(F) = \alpha F$ to the expression of $\bar{A}(F)$, we have:

$$\begin{aligned}
\bar{A}(F) &= \frac{T(F) \cdot \pi}{T(F) \cdot \pi + F \cdot (1 - \pi)} \\
&= \frac{\alpha F \cdot \pi}{\alpha F \cdot \pi + F \cdot (1 - \pi)} \\
&= \frac{\alpha \cdot \pi}{\alpha \cdot \pi + (1 - \pi)}
\end{aligned}$$

□

*Proof.* For the first part of Claim 1, we simply extend the ROC curve $T(F')$ to equal zero for all $F \leq F(B)$, such that Lemma 1 implies $\bar{A}(F) = 0$ for these $F$. Because $C(F)$ is monotonously falling in $F$, we get $\lim_{C \to C_{\max}} \text{ACC}(C) = 0$.

For the concave case of Claim 1, we extend the ROC curve linearly for $F < F(B)$ using the line connecting the origin to the point $(F(B), T(F(B)))$. This yields a concave ROC curve: If it did not, the slope of the newly added segment had to be too small for concavity to hold. We claim that in this case, the known part of the ROC was not extendable to a concave ROC: A slope larger than the proposed one on any sub-segment would have $T(F)$ hit the $x-$axis before the origin. This would have forced the ROC curve to equal zero in an interval around $F = 0$, precluding concavity because $T(1) = 1$.

Now, the extended curve has slope $\alpha(F(B)) = \frac{T(F(B))}{F(B)}$, around the origin, and applying Proposition 2 yields

$$\lim_{C \to C_{\max}} A(C) = \frac{\alpha(F(B))\pi}{\alpha(F(B))\pi + (1 - \pi)} = A(B),$$

where the second equality uses Lemma 1.

For claim 2, we note that $F(B) > 0$ because $B < \lim_{F \to 0} C(F)$. Meanwhile $T(F(B)) > 0$ because $A(B) = \bar{A}(F(B)) > 0$. Thus, we simply construct the ROC curve by connecting $(F(B), T(F(B)))$ to $(0, T(F(B)))$ by a horizontal line. The resulting ROC curve thus has $T(0) > 0$ and by Proposition 2, a verifier with this ROC curve achieves $\lim_{C \to C(0)} A(C) = 1$.

If we assume the ROC curve to be concave, we know that its right-sided derivatives exist and are non-increasing Rockafellar (1970). We then linearly extend the ROC curve from $(\mathrm{F}(B), \mathrm{T}(\mathrm{F}(B)))$ with slope equal to the right-sided derivative $\mathrm{T}'_+(\mathrm{F}(B))$ at $\mathrm{F}(B)$ to obtain the largest possible concave extension of the ROC curve. There are now three cases: First, this extension hits the $x-$ axis before the origin. Because $T(0) \geq 0$, this does not yield a valid ROC curve, showing that the observed scaling has not been compatible with a concave ROC to begin with. Second, we get $T(0) > 0$. In that case, we obtain $\lim_{C \to C(0)} A(C) = 1$ by Proposition 2. In the last case, we get $T(0) = 0$ and thus $\lim_{C \to C(0)} A(C) < 1$. We claim that this can only happen if the right one-sided derivative of $\bar{A}$ vanishes at $\mathrm{F}(B)$: For $T(0) = 0$ to happen, we need

$$\lim_{x \to ^+ \mathrm{F}(B)} \frac{\mathrm{T}(x) - \mathrm{T}(\mathrm{F}(B))}{x - \mathrm{F}(B)} = \mathrm{T}'_+(\mathrm{F}(B)) = \frac{\mathrm{T}(\mathrm{F}(B))}{\mathrm{F}(B)}.$$

But with this, we get

$$\lim_{x \to ^+ \mathrm{F}(B)} \frac{\bar{A}(x) - \bar{A}(\mathrm{F}(B))}{x - \mathrm{F}(B)}$$

$$= \lim_{x \to ^+ \mathrm{F}(B)} \frac{\frac{T(x)\pi}{T(x)\pi + x(1-\pi)} - \frac{T(\mathrm{F}(B))\pi}{T(\mathrm{F}(B))\pi + \mathrm{F}(B)(1-\pi)}}{x - \mathrm{F}(B)}$$

$$= \lim_{x \to ^+ \mathrm{F}(B)} \frac{T(x)\pi(T(\mathrm{F}(B))\pi + \mathrm{F}(B)(1-\pi)) - T(\mathrm{F}(B))\pi(T(x)\pi + x(1-\pi))}{(x - \mathrm{F}(B))(T(\mathrm{F}(B))\pi + \mathrm{F}(B)(1-\pi))(T(x)\pi + x(1-\pi))}$$

$$= \lim_{x \to ^+ \mathrm{F}(B)} \frac{T(x)\pi \mathrm{F}(B)(1-\pi) - T(\mathrm{F}(B))\pi x(1-\pi)}{(x - \mathrm{F}(B))(T(\mathrm{F}(B))\pi + \mathrm{F}(B)(1-\pi))(T(x)\pi + x(1-\pi))}$$

$$= (1-\pi)\pi \lim_{x \to ^+ \mathrm{F}(B)} \frac{T(x)\mathrm{F}(B) - T(\mathrm{F}(B))x}{(x - \mathrm{F}(B))(T(\mathrm{F}(B))\pi + \mathrm{F}(B)(1-\pi))(T(x)\pi + x(1-\pi))}$$

$$= (1-\pi)\pi \lim_{x \to ^+ \mathrm{F}(B)} (T(\mathrm{F}(B))\pi + \mathrm{F}(B)(1-\pi))(T(x)\pi + x(1-\pi))$$

$$\cdot \lim_{x \to ^+ \mathrm{F}(B)} \frac{T(x)\mathrm{F}(B) - T(\mathrm{F}(B))x}{(x - \mathrm{F}(B))}$$

$$= (1-\pi)\pi(T(\mathrm{F}(B))\pi + \mathrm{F}(B)(1-\pi))^2$$

$$\cdot \lim_{x \to ^+ \mathrm{F}(B)} \frac{T(x)\mathrm{F}(B) - T(\mathrm{F}(B))x}{(x - \mathrm{F}(B))}$$

$$= (1-\pi)\pi(T(\mathrm{F}(B))\pi + \mathrm{F}(B)(1-\pi))^2$$

$$\cdot \lim_{x \to ^+ \mathrm{F}(B)} \frac{F(B)(T(x) - T(F(B))) - (x - F(B))T(F(B))}{(x - \mathrm{F}(B))}$$

$$= (1-\pi)\pi(T(\mathrm{F}(B))\pi + \mathrm{F}(B)(1-\pi))^2$$

$$\cdot \lim_{x \to ^+ \mathrm{F}(B)} \frac{F(B)(T(x) - T(F(B))) - (x - F(B))T(F(B))}{(x - \mathrm{F}(B))}$$

$$= (1-\pi)\pi(T(\mathrm{F}(B))\pi + \mathrm{F}(B)(1-\pi))^2(T(F(B)) - T(F(B)))$$

$$= (1-\pi)\pi(T(\mathrm{F}(B))\pi + \mathrm{F}(B)(1-\pi))^2(0) = 0$$

By the calculations at the start of the proof of Proposition 1, this implies that the left side derivative of $\mathrm{ACC}(C)$ has to vanish as well.

$\square$

## C.6   PROOF OF PROPOSITION 4

*Proof.* We will first prove the integral formula, and prove the monotonicity statement, as

**Proposition 9.** *Let $f$ be a verifier with concave ROC curve $T(F)$ for which Assumption 1 holds. Then BoN accuracy $ACC(g_N)$ is non-decreasing in $N$.*

later in Appendix C.11.

For the integral formula, we first notice that $H(k, p)$ can be written as an integral involving the ROC curve:

**Lemma 2.** *Let $f$ be a verifier for which Assumption 1 holds, and $T : [0, 1] \mapsto [0, 1]$ be the ROC curve induced by $f$. Then we have*

$$H(k, p) = \begin{cases} k \int_0^1 (1 - (1 - T(F))^p)(1 - F)^{k-1} dF & \text{if } k > 0 \\ 1 & \text{if } k = 0 \end{cases}.$$

*Proof.* We prove the lemma for the case of absolutely continuous scores $f(x)$ with a density function. The extension to discrete scores $f(x)$ will be discussed later in Appendix D.

Let $S_N = \{x_1, \cdots, x_N\}$ be a set of $N$ iid samples from the generator. We analyze the probability that the selected output under Best-of-$N$ sampling is correct, conditional on exactly $p$ of the $N$ samples being positive (i.e., $y(x_i) = 1$ for $p$ values of $i$). Denote $k = N - p$ as the total number of negative samples. Define $S_1^+, \ldots, S_p^+$ as i.i.d. draws from the score distribution $f(x)$ conditioned on $y(x) = 1$, and $S_1^-, \ldots, S_k^-$ analogously for $y(x) = 0$.

Then the conditional accuracy given $p$ positives is:

$$\mathbb{E}\left[ y\left( \arg\max_{i \in [N]} f(x_i) \right) \,\middle|\, \sum y(x_i) = p \right] = \Pr\left( \max_{i \in [p]} S_i^+ > \max_{j \in [k]} S_j^- \right).$$

To compute this, observe that:

$$\Pr\left( \max_{i \in [p]} S_i^+ > z \right) = 1 - \Pr(S^+ \leq z)^p, \quad \Pr\left( \max_{j \in [k]} S_j^- \leq z \right) = \Pr(S^- \leq z)^k.$$

Assuming the density $p_{S^-}(z)$ exists, the density of $\max_j S_j^-$ is:

$$p_{\max S^-}(z) = k \cdot \Pr(S^- \leq z)^{k-1} \cdot p_{S^-}(z).$$

Thus the conditional probability becomes:

$$\Pr\left( \max_{i \in [p]} S_i^+ > \max_{j \in [k]} S_j^- \right)$$

$$= \int \Pr\left( \max S^+ > z \right) \cdot p_{\max S^-}(z) \, dz$$

$$= k \int \left( 1 - \Pr(S^+ \leq z)^p \right) \cdot \Pr(S^- \leq z)^{k-1} \cdot p_{S^-}(z) \, dz.$$

We use the ROC-based expressions:

$$\text{FNR}(z) = \Pr(S^+ \leq z), \text{TNR}(z) = \Pr(S^- \leq z), \text{F}(z) = 1 - \text{TNR}(z), \text{T}(\text{F}(z)) = 1 - \text{FNR}(z),$$

and change variables by setting $u = \text{TNR}(z) = 1 - \text{F}(z)$. Since $\frac{du}{dz} = -p_{\text{F}}(z)$, the change of variables gives:

$$k \int \left( 1 - (1 - \text{T}(\text{F}(z)))^p \right) (1 - \text{F}(z))^{k-1} p_{S^-}(z) \, dz$$

$$= k \int_0^1 \left( 1 - (1 - \text{T}(\text{F}))^p \right) (1 - \text{F})^{k-1} \, d\text{F}.$$

This final integral matches the definition of $H(k, p)$, and averaging over $p \sim \text{Bin}(N, \pi)$ yields the desired result:

$$\text{ACC}(g_N^f) = \mathbb{E}_{p \sim \text{Bin}(N, \pi)}[H(N - p, p)].$$

$\square$

We now set $a(\mathrm{F}) := (1 - \mathrm{F})(1 - \pi)$, $b(\mathrm{F}) := \pi(1 - \mathrm{T(F)})$ such that $\psi(F) = a(F) + b(F)$. We then simplify

$$
\begin{aligned}
\mathrm{ACC}(g_n) &= \mathbb{E}_{p \sim B(\pi, n)} H(n - p, p) && \text{(change of variable } k = n - p) \\
&= \mathbb{E}_{k \sim B((1-\pi), n)} H(k, n - k) \\
&= \sum_{k=0}^{n} \binom{n}{k} (\pi)^{n-k} (1 - \pi)^k H(k, n - k) \\
&= n\pi^n + \sum_{k=1}^{n} \binom{n}{k} (\pi)^{n-k} (1 - \pi)^k k \int_0^1 (1 - (1 - \mathrm{T(F)})^{n-k})(1 - \mathrm{F})^{k-1} d\mathrm{F} \\
&= n\pi^n + \sum_{k=1}^{n} \binom{n}{k} (\pi)^{n-k} (1 - \pi)^k \left( 1 - k \int_0^1 ((1 - \mathrm{T(F)})^{n-k} (1 - \mathrm{F})^{k-1} d\mathrm{F} \right) \\
&= n\pi^n + 1 - n\pi^n - \sum_{k=1}^{n} \binom{n}{k} (\pi)^{n-k} (1 - \pi)^k k \int_0^1 ((1 - \mathrm{T(F)})^{n-k} (1 - \mathrm{F})^{k-1} d\mathrm{F} \\
&= 1 - \int_0^1 \sum_{k=1}^{n} k \binom{n}{k} (\pi)^{n-k} (1 - \pi)^k ((1 - \mathrm{T(F)})^{n-k} (1 - \mathrm{F})^{k-1} d\mathrm{F} \\
&= 1 - (1 - \pi) \int_0^1 \sum_{k=1}^{n} k \binom{n}{k} (\pi)^{n-k} (1 - \pi)^{k-1} ((1 - \mathrm{T(F)})^{n-k} (1 - \mathrm{F})^{k-1} d\mathrm{F} \\
&= 1 - (1 - \pi) \int_0^1 \sum_{k=1}^{n} k \binom{n}{k} b(\mathrm{F})^{n-k} a(\mathrm{F})^{k-1} d\mathrm{F} \\
&= 1 - (1 - \pi) n \int_0^1 \sum_{k=1}^{n} \frac{k}{n} \binom{n}{k} b(\mathrm{F})^{n-k} a(\mathrm{F})^{k-1} d\mathrm{F} \\
&= 1 - (1 - \pi) n \int_0^1 \sum_{k=1}^{n} \binom{n-1}{k-1} b(\mathrm{F})^{n-k} a(\mathrm{F})^{k-1} d\mathrm{F} && (\tfrac{k}{n}\binom{n}{k} = \binom{n-1}{k-1}) \\
&= 1 - (1 - \pi) n \int_0^1 \sum_{k=0}^{n-1} \binom{n-1}{k} b(\mathrm{F})^{n-k-1} a(\mathrm{F})^{k} d\mathrm{F} \\
&\qquad\qquad \text{(binomial expansion: } \sum_{k=0}^{n-1} \binom{n-1}{k} b^{n-1-k} a^k = (a + b)^{n-1}) \\
&= 1 - (1 - \pi) n \int_0^1 (b(\mathrm{F}) + a(\mathrm{F}))^{n-1} d\mathrm{F} \\
&= 1 - (1 - \pi) n \int_0^1 (\psi(F))^{n-1} \, d\mathrm{F}.
\end{aligned}
$$

$\square$

### C.7 PROOF OF THEOREM 1

*Proof.* We divide the proof into two cases: Case 1, when $\mathrm{T}(0) > 0$, and Case 2, when $\mathrm{T}(0) = 0$.

**Case 1:** $\mathrm{T}(0) > 0$. In this case, there is a threshold $\tau$, such that $\mathrm{F}(\tau) = 0$ while $\mathrm{T}(\tau) > 0$. This implies that all outputs $x$ with $f(x) \geq \tau$ are true positives ($y(x) = 1$) and that at least one such $x$ has positive probability $\mathrm{Pr}_{y \sim g}[y = x] =: c > 0$. This means that whenever our $N$ samples contain one of these $x$, Best-of-N will return a correct answer with $y(x) = 1$. But the probability that none of $N$ independent samples contains one of these $x$ is at most $(1 - c)^N$, which decays to zero exponentially.

**Case 2:** $\mathrm{T}(0) = 0$. In this case, our objective is to show that the asymptotic performance of the Best-of-$N$ strategy is determined by the slope of the ROC curve $\mathrm{T(F)}$ near the origin.

*Proof sketch:* Recall that the expected performance is given by $\mathbb{E}[H(n-p, p)]$, where $p \sim \mathrm{Bin}(n, \pi)$. This expression is difficult to analyze directly due to the randomness of $p$, so our strategy is to

approximate it using the deterministic surrogate $H(\mathbb{E}[n-p], \mathbb{E}[p]) = H((1-\pi)n, \pi n)$ and then argue in the limit ($n \to \infty$), these two expressions converge to the same value, namely

$$\lim_{n \to \infty} \mathbb{E}[H(n-p, p)] = \lim_{n \to \infty} H(\mathbb{E}[n-p], \mathbb{E}[p]) = \frac{T'(0)\pi}{T'(0)\pi + 1 - \pi}$$

We now proceed to the complete proof. Recall the expression for $H(k, p)$ as:

$$H(k, p) = k \int_0^1 \left(1 - (1 - T(F))^p\right)(1 - F)^{k-1} \, dF$$

$$= k \int_0^1 (1 - F)^{k-1} \, dF - k \int_0^1 (1 - T(F))^p (1 - F)^{k-1} \, dF$$

$$= 1 - k \int_0^1 (1 - T(F))^p (1 - F)^{k-1} \, dF$$

Now consider $H(n - p, p)$ with $p \sim \text{Bin}(n, \pi)$. Since $\mathbb{E}[p] = n\pi$, we have:

$$H(\mathbb{E}[n-p], \mathbb{E}[p]) = 1 - n(1 - \pi) \int_0^1 (1 - T(F))^{n\pi}(1 - F)^{n(1-\pi)-1} \, dF.$$

We first show that when $n \to \infty$, the deterministic approximation $H(\mathbb{E}[n-p], \mathbb{E}[p])$ converges to a closed-form expression that depends only on the initial slope of the ROC curve $T'(0)$ and the class prior $\pi$:

**Lemma 3.** *Let $T : [0, 1] \to [0, 1]$ denote the true positive rate as a function of the false positive rate (i.e., the ROC curve), and assume $T(0) = 0$ and that $T$ is differentiable at 0. Then,*

$$\lim_{n \to \infty} H(\mathbb{E}[n-p], \mathbb{E}[p]) = \frac{T'(0)\pi}{T'(0)\pi + 1 - \pi},$$

*where $T'(0)$ denotes the derivative of the ROC curve at the origin (i.e., its initial slope), and $\pi \in (0, 1)$ is the class prior probability of a positive instance.*

*Proof.* To analyze the limit of $H(\mathbb{E}[n-p], \mathbb{E}[p])$, we first perform a change of variables. Let $u = n(1 - \pi)F$, so that $du = n(1 - \pi) \, dF$, and rewrite the integral as:

$$H(\mathbb{E}[n-p], \mathbb{E}[p]) = 1 - n(1 - \pi) \int_0^1 (1 - T(F))^{n\pi}(1 - F)^{n(1-\pi)-1} \, dF$$

$$= 1 - \int_0^{n(1-\pi)} \left(1 - T\left(\frac{u}{n(1-\pi)}\right)\right)^{n\pi} \left(1 - \frac{u}{n(1-\pi)}\right)^{n(1-\pi)-1} \, du.$$

We extend the upper limit of the integral to infinity by introducing an indicator function:

$$H(\mathbb{E}[n-p], \mathbb{E}[p])$$

$$= 1 - \int_0^\infty \left(1 - T\left(\frac{u}{n(1-\pi)}\right)\right)^{n\pi} \left(1 - \frac{u}{n(1-\pi)}\right)^{n(1-\pi)-1} \mathbb{I}(u \le n(1-\pi)) \, du.$$

To justify exchanging the limit and the integral as $n \to \infty$, we apply the Dominated Convergence Theorem. The indicator function is bounded above by 1, and for the other terms, we use the inequality $1 - x \le e^{-x}$:

$$\left(1 - T\left(\frac{u}{n(1-\pi)}\right)\right)^{n\pi} \le \exp\left(-n\pi T\left(\frac{u}{n(1-\pi)}\right)\right) \le 1,$$

$$\left(1 - \frac{u}{n(1-\pi)}\right)^{n(1-\pi)-1} \le \exp\left(-\frac{(n(1-\pi)-1)u}{n(1-\pi)}\right).$$

The product of the two terms is thus dominated by an exponential function with a negative exponent, which is integrable over $u \in [0, \infty)$. Thus, we may take the limit inside the integral:

$$\lim_{n \to \infty} H(\mathbb{E}[n-p], \mathbb{E}[p]) = 1 - \int_0^\infty \lim_{n \to \infty} \left(1 - \mathrm{T}\left(\frac{u}{n(1-\pi)}\right)\right)^{n\pi} \left(1 - \frac{u}{n(1-\pi)}\right)^{n(1-\pi)-1} du.$$

We now compute the pointwise limit of the integrand. The key term is:

$$\lim_{n \to \infty} \left(1 - \mathrm{T}\left(\frac{u}{n(1-\pi)}\right)\right)^{n\pi}.$$

**Claim 1.** *Let T be differentiable at 0 with $T(0) = 0$. Then:*

$$\lim_{n \to \infty} \left(1 - T\left(\frac{u}{n(1-\pi)}\right)\right)^{n\pi} = \exp\left(-\frac{\pi u}{1-\pi} T'(0)\right).$$

*Proof.* We rewrite using the exponential:

$$\left(1 - \mathrm{T}\left(\frac{u}{n(1-\pi)}\right)\right)^{n\pi} = \exp\left(n\pi \log\left(1 - \mathrm{T}\left(\frac{u}{n(1-\pi)}\right)\right)\right).$$

Letting $t = \frac{1}{n}$ and applying L'Hôpital's Rule:

$$\lim_{n \to \infty} n\pi \log\left(1 - \mathrm{T}\left(\frac{u}{n(1-\pi)}\right)\right) = \lim_{t \to 0} \frac{\pi \log\left(1 - \mathrm{T}\left(\frac{tu}{1-\pi}\right)\right)}{t} \quad \text{(change of variable } t = \tfrac{1}{n})$$

$$= \pi \cdot \left(\lim_{t \to 0} \frac{d}{dt} \log\left(1 - \mathrm{T}\left(\frac{tu}{1-\pi}\right)\right)\right) \quad \text{(L'Hôpital's Rule)}$$

$$= \lim_{t \to 0} -\pi \frac{u\mathrm{T}'(\frac{tu}{1-\pi})}{(1-\pi)(1 - \mathrm{T}(\frac{tu}{1-\pi}))}$$

$$= \lim_{t \to 0} -\pi \frac{u\mathrm{T}'(\frac{tu}{1-\pi})}{(1-\pi)}$$

$$= -\pi \frac{u\mathrm{T}'(0)}{(1-\pi)}$$

$$= -\frac{\pi u}{1-\pi} \mathrm{T}'(0)$$

$$\square$$

The second term converges similarly:

$$\left(1 - \frac{u}{n(1-\pi)}\right)^{n(1-\pi)-1} \to e^{-u}.$$

Putting it all together:

$$\lim_{n \to \infty} H(\mathbb{E}[n-p], \mathbb{E}[p]) = 1 - \int_0^\infty e^{-\frac{\pi u}{1-\pi} \mathrm{T}'(0)} \cdot e^{-u} \, du$$

$$= 1 - \int_0^\infty \exp\left(-\left(\frac{\pi}{1-\pi} \mathrm{T}'(0) + 1\right) u\right) du \quad \text{(Claim 1)}$$

$$= 1 - \frac{1}{\frac{\pi}{1-\pi} \mathrm{T}'(0) + 1}$$

$$= \frac{\mathrm{T}'(0)\pi}{\mathrm{T}'(0)\pi + 1 - \pi}.$$

$$\square$$

To complete the proof, it remains to show that

$$\lim_{n\to\infty} \mathbb{E}(H(n-p,p)) = \lim_{n\to\infty} H(\mathbb{E}[n-p], \mathbb{E}[p])$$

We approach this by conditioning on the event that the random variable $p \sim \text{Bin}(n, \pi)$ concentrates near its expectation. Fix an arbitrary $t > 0$, and define the high-probability event

$$E_t := \left\{ \left| \frac{p - \mathbb{E}[p]}{n} \right| < t \right\},$$

which corresponds to the event that the empirical frequency of positive samples deviates from its mean by less than $t$. Conditioned on this event, we can decompose the expectation in the following way:

$$\mathbb{E}[H(n-p,p)] = \Pr(E_t) \cdot \mathbb{E}[H(n-p,p) \mid E_t] + \Pr(\neg E_t) \cdot \mathbb{E}[H(n-p,p) \mid \neg E_t].$$

To proceed, we use the following lemma, which shows that the success probability $H(n-p,p)$ increases with the number of positive samples $p$:

**Lemma 4.** *For fixed $n$, $H(n-p,p) = 1 - (n-p) \int_0^1 (1 - T(F))^p (1-F)^{n-p-1} dF$ increases in $p$.*

*Proof.* For integer values of $p$, this follows from the definition of

$$H(n-p,p) = \Pr\left( \max_{i\in[p]} S_i^+ > \max_{j\in[n-p]} S_j^- \right)$$

$$\leq \Pr\left( \max_{i\in[p+1]} S_i^+ > \max_{j\in[n-p]} S_j^- \right)$$

$$\leq \Pr\left( \max_{i\in[p+1]} S_i^+ > \max_{j\in[n-p-1]} S_j^- \right)$$

by strict inclusion of the events in the corresponding probabilities.

For non-integer values of $p$, we define the random variables $S_\varepsilon^+$ and $S_\varepsilon^-$ via

$$\Pr(S_\varepsilon^\pm \geq z) = \Pr(S^\pm \geq z)^\varepsilon,$$

and assume them to be independent from the $S_i^\pm$. Then

$$\Pr\left( \max\{\max_{i\in[p]} S_i^+, S_\varepsilon^+\} > z \right) = 1 - \Pr(S^+ \leq z)^{p+\varepsilon},$$

$$\Pr\left( \max\{\max_{j\in[n-p]} S_j^-, S_\varepsilon^-\} \leq z \right) = \Pr(S^- \leq z)^{n-p+\varepsilon}.$$

Assuming the density $p_{S^-}(z)$ exists, the density of $\mathbf{S}_{n-p+\varepsilon}^- =: \max\{\max_{j\in[n-p]} S_j^-, S_\varepsilon^-\}$ thus equals:

$$p_{\max \mathbf{S}_{n-p+\varepsilon}^-}(z) = (n-p+\varepsilon) \cdot \Pr(S^- \leq z)^{n-p-1+\varepsilon} \cdot p_{S^-}(z).$$

Repeating the steps from Lemma 2, we get

$$H(n-p-\varepsilon, p+\varepsilon) = \Pr\left( \max\{\max_{i\in[p]} S_i^+, S_\varepsilon^+\} > \max\{\max_{j\in[n-p-1]} S_j^-, S_{1-\varepsilon}^-\} \right)$$

for any positive integer $p$ and $\varepsilon \in (0,1)$. For any positive integers $p, q$ such that $p + q < n$ and $\varepsilon \in (0, 1)$, we then have

$$H(n-p-q-\varepsilon, p+q+\varepsilon) = \Pr\left( \max\{\max_{i\in[p+q]} S_i^+, S_\varepsilon^+\} > \max\{\max_{j\in[n-p-q-1]} S_j^-, S_{1-\varepsilon}^-\} \right)$$

$$\geq \Pr\left( \max_{i\in[p+q]} S_i^+ > \max_{j\in[n-p-q]} S_j^- \right)$$

$$\geq \Pr\left( \max_{i\in[p]} S_i^+ > \max_{j\in[n-p]} S_j^- \right)$$

$$= H(n-p,p),$$

where the second step uses that we can couple $S^-_{n-p-q-\varepsilon}$ and $S^-_{n-p-q}$ such that the latter is larger with probability one, due to its CDF being larger at any point. An analogous argument works for $p - q - \varepsilon$, establishing monotonicity. $\square$

Recall the decomposition of the expectation as:

$$\mathbb{E}[H(n - p, p)] = \mathbb{E}[H(n - p, p) \mid E_t] \cdot \mathbb{P}(E_t) + \mathbb{E}[H(n - p, p) \mid \neg E_t] \cdot \mathbb{P}(\neg E_t).$$

By Hoeffding's inequality, $\mathbb{P}(E_t) \to 1$ as $n \to \infty$ for any fixed $t > 0$. Since $H(n - p, p) \in [0, 1]$, the second term vanishes in the limit. Hence, for any $t > 0$

$$\limsup_{n \to \infty} \mathbb{E}[H(n - p, p)] = \limsup_{n \to \infty} \mathbb{E}[H(n - p, p) \mid E_t]$$

and

$$\liminf_{n \to \infty} \mathbb{E}[H(n - p, p)] = \liminf_{n \to \infty} \mathbb{E}[H(n - p, p) \mid E_t].$$

Now, fix $t > 0$ and consider the conditional expectation over $E_t = \left\{ \left| \frac{p - n\pi}{n} \right| < t \right\}$. Since $H(n - p, p)$ is monotone increasing in $p$ (by Lemma 4), on $E_t$ we always have:

$$H(n(1 - \pi + t), n(\pi - t)) \le H(n - p, p) \le H(n(1 - \pi - t), n(\pi + t)).$$

Therefore,

$$H(n(1 - \pi + t), n(\pi - t)) \le \mathbb{E}[H(n - p, p) \mid E_t] \le H(n(1 - \pi - t), n(\pi + t)).$$

This means that

$$
\begin{aligned}
\limsup_{n \to \infty} \mathbb{E}[H(n - p, p)] &= \limsup_{n \to \infty} \mathbb{E}[H(n - p, p) \mid E_t] \\
&\le \limsup_{n \to \infty} H(n(1 - \pi - t), n(\pi + t)) \\
&= \lim_{n \to \infty} H(n(1 - \pi - t), n(\pi + t)) \\
&= \frac{\mathrm{T}'(0)(\pi + t)}{\mathrm{T}'(0)(\pi + t) + 1 - (\pi + t)}
\end{aligned}
$$

$$
\begin{aligned}
\liminf_{n \to \infty} \mathbb{E}[H(n - p, p)] &= \liminf_{n \to \infty} \mathbb{E}[H(n - p, p) \mid E_t] \\
&\ge \liminf_{n \to \infty} H(n(1 - \pi + t), n(\pi - t)) \\
&= \lim_{n \to \infty} H(n(1 - \pi + t), n(\pi - t)) \\
&= \frac{\mathrm{T}'(0)(\pi - t)}{\mathrm{T}'(0)(\pi - t) + 1 - (\pi - t)}
\end{aligned}
$$

But as $\frac{\mathrm{T}'(0)(\pi)}{\mathrm{T}'(0)(\pi) + 1 - (\pi)}$ is continuous in $\pi$ and $t > 0$ was chosen arbitrarily, we get

$$\limsup_{n \to \infty} \mathbb{E}[H(n - p, p)] \le \frac{\mathrm{T}'(0)\pi}{\mathrm{T}'(0)\pi + 1 - \pi} \le \liminf_{n \to \infty} \mathbb{E}[H(n - p, p)]$$

and thus

$$\lim_{n \to \infty} \mathbb{E}[H(n - p, p)] = \frac{\mathrm{T}'(0)\pi}{\mathrm{T}'(0)\pi + 1 - \pi}$$

$\square$

### C.8 PROOF OF PROPOSITION 7

We make use of the following Lemma.

**Lemma 5** (Layer-cake reformulation for $\psi(F)^{N-1}$). *Let $\psi : [0, 1] \to [0, 1]$ be a convex and decreasing function and $N \ge 2$. Set $M = \max_F \psi(F)$. Then:*

$$\int_0^1 \psi(F)^{N-1} \, dF = (N - 1) \int_0^M \psi^{-1}(a) \cdot a^{N-2} \, da.$$

*Proof of Lemma 5.* Since $\psi$ is convex and decreasing, it is continuous on $(0,1]$. Furthermore, as the exact value of $\psi(0)$ does not affect the integral, we can without loss of generality assume that $\psi$ is continuous on the whole interval $[0,1]$. This means that the inverse $\psi^{-1}$ is a continuous function defined on $[0,M]$. We now begin by applying the *layer-cake representation* to the non-negative function $\psi(F)^{N-1}$. Since $\psi(F)$ is decreasing and bounded in $[0,1]$, we can write:

$$\int_0^1 \psi(F)^{N-1}\,dF = \int_0^1 \mathbb{P}[\psi(F)^{N-1} \geq a]\,da$$

$$= \int_0^{M^{N-1}} \mathbb{P}[\psi(F)^{N-1} \geq a]\,da + \int_{M^{N-1}}^1 \mathbb{P}[\psi(F)^{N-1} \geq a]\,da$$

$$= \int_0^{M^{N-1}} \psi^{-1}(a^{1/(N-1)})\,da + 0.$$

The last step uses that for the left term $\psi(F)^{N-1} \geq a$ is equivalent to $F \leq \psi^{-1}(a^{1/(N-1)})$ since $\psi$ is decreasing. For the right term, we use that $\psi(F)^{N-1} \leq M^{N-1}$ with probability one.

Now we perform a change of variables. Let $u = a^{1/(N-1)}$, so that $a = u^{N-1}$ and $da = (N-1)u^{N-2}\,du$. Then:

$$\int_0^{M^{N-1}} \psi^{-1}(a^{1/(N-1)})\,da = \int_0^M \psi^{-1}(u) \cdot (N-1)u^{N-2}\,du$$

$$= (N-1)\int_0^M \psi^{-1}(u) \cdot u^{N-2}\,du.$$

$\square$

We now prove Proposition 7 using Lemma 5:

*Proof of Proposition 7.* By Proposition 4, we have

$$\mathrm{ACC}(g_N) = 1 - (1-\pi)N\int_0^1 \psi(F)^{N-1}dF,$$

where $\psi(F) = (1-\pi)(1-F) + \pi(1 - T(F))$ is convex and decreasing with $\psi(1) = 0$ and $\psi(0) = 1 - \pi T(0)$. Lemma 5 then gives us

$$\mathrm{ACC}(g_N) = 1 - (1-\pi)N\int_0^1 \psi(F)^{N-1}dF$$

$$= 1 - (1-\pi)N(N-1)\int_0^{1-\pi T(0)} \psi^{-1}(a) \cdot a^{N-2}dF.$$

$\square$

## C.9 Proof of Proposition 5

*Proof.* Fix $N \in \mathbb{N}$. By definition, we have

$$C(F_N) = \frac{1}{\mathrm{T}(F_N)\pi + \mathrm{F}_N(1-\pi)} = N.$$

Then the expected accuracy of rejection sampling is

$$\mathrm{ACC}(g^{h_{F_N}}) = N \cdot \pi \cdot \mathrm{T}(F_N).$$

By Proposition 7, due to concavity of the ROC, BoN accuracy is given by:

$$\mathrm{ACC}(g_N) = 1 - (1-\pi)(N^2 - N)\int_0^{1-\pi T(0)} \psi^{-1}(a)a^{N-2}da,$$

where $\psi(F) = (1 - \pi)(1 - F) + \pi(1 - \mathrm{T}(F))$ is the probability that a sample is rejected.

To compare the two accuracies, we rewrite the inequality to be shown:

$$N \cdot \pi \cdot \mathrm{T}(F_N) \geq 1 - (1 - \pi)(N^2 - N) \int_0^{1 - \pi T(0)} \psi^{-1}(a) a^{N-2} da.$$

Since we know that:

$$\pi \cdot \mathrm{T}(F_N) + (1 - \pi) \cdot F_N = \frac{1}{N},$$

we can rewrite the left-hand side:

$$N \cdot \pi \cdot \mathrm{T}(F_N) = 1 - N(1 - \pi)F_N.$$

So it is sufficient to prove:

$$1 - N(1 - \pi)F_N \geq 1 - (1 - \pi)N(N - 1) \int_0^{1 - \pi T(0)} \psi^{-1}(a) a^{N-2} da,$$

which is equivalent to proving:

$$F_N \leq (N - 1) \int_0^{1 - \pi T(0)} \psi^{-1}(a) a^{N-2} da.$$

We rewrite $F_N$ in terms of $\psi$. Recall that

$$C(F_N) = \frac{1}{\mathrm{T}(F_N)\pi + F_N(1 - \pi)} = N.$$

Using the identity $\psi(F) = 1 - (\mathrm{T}(F)\pi + \mathrm{F}(1 - \pi))$, we obtain:

$$\psi(F_N) = 1 - \frac{1}{N} = \frac{N - 1}{N}, \quad \text{so} \quad F_N = \psi^{-1}\left(\frac{N - 1}{N}\right).$$

In particular, this implies that $\psi^{-1}(x)$ is defined for $0 \leq x \leq \frac{N-1}{N}$ and it suffices to show:

$$\psi^{-1}\left(\frac{N - 1}{N}\right) \leq (N - 1) \int_0^{1 - \pi T(0)} \psi^{-1}(a) a^{N-2} da.$$

We observe that the right hand side nearly matches the expected value of $\hat{\psi}^{-1}(a)$ under a Beta distribution with parameters $(N - 1, 1)$:

**Lemma 6** (Beta expectation form of layer-cake integral). *Let $f : [0, 1] \to [0, 1]$ be a function, and let $N \geq 2$. Then:*

$$(N - 1) \int_0^1 f(a) \cdot a^{N-2} \, da = \mathbb{E}_{a \sim \mathrm{Beta}(N-1,1)} [f(a)].$$

*Proof.* Recall that the probability density function of the $\mathrm{Beta}(\alpha, \beta)$ distribution on $[0, 1]$ is given by

$$p_a(x) = \frac{1}{\mathrm{B}(\alpha, \beta)} x^{\alpha-1}(1 - x)^{\beta-1}, \quad \text{for } x \in [0, 1],$$

where $\mathrm{B}(\alpha, \beta)$ is the Beta function.

For the case $\alpha = N - 1$ and $\beta = 1$, since $\mathrm{B}(N - 1, 1) = \frac{1}{N-1}$, we have:

$$p_a(x) = (N - 1)x^{N-2}, \quad x \in [0, 1]$$

Therefore, the expectation of $f(a)$ under $a \sim \mathrm{Beta}(N - 1, 1)$ is:

$$\mathbb{E}_{a \sim \mathrm{Beta}(N-1,1)} [f(a)] = \int_0^1 f(a) \cdot (N - 1)a^{N-2} \, da.$$

$\square$

Now, since that the ROC curve $T(F)$ is concave, it follows that $\psi(F)$ is convex, and hence $\psi^{-1}$ is convex on its image. We extend the domain by setting

$$\hat{\psi}^{-1}(a) = \begin{cases} \psi^{-1}(a) & \text{if } a \leq M \\ 0 & \text{if } a > M \end{cases}.$$

The extended function $\hat{\psi}^{-1}$ remains convex on $[0, 1]$ because $\psi^{-1}$ was decreasing with value $0$ at the right end of its domain.

We can therefore apply Jensen's inequality to obtain:

$$\psi^{-1}\left(\frac{N-1}{N}\right) = \hat{\psi}^{-1}\left(\frac{N-1}{N}\right) = \hat{\psi}^{-1}\left(\mathbb{E}_{a \sim \text{Beta}(N-1,1)}[a]\right)$$

$$\leq \mathbb{E}_{a \sim \text{Beta}(N-1,1)}[\hat{\psi}^{-1}(a)] = (N-1)\int_0^1 \hat{\psi}^{-1}(a)a^{N-2}da = (N-1)\int_0^{1-\pi T(0)} \psi^{-1}(a)a^{N-2}da.$$

This concludes the proof. $\qquad\square$

## C.10 Proof of Theorem 2

*Proof.* We begin by considering alternative equivalent characterizations of concave ROC curves:

**Claim 2.** *Let $T : [0, 1] \to [0, 1]$ be an ROC curve for a verifier $f$. Then the following are equivalent:*

(a) *$T$ is piecewise linear, strictly increasing, and concave with $T(1) = 1$, and for all $F \in [0, 1]$,*

$$F \leq T(F) \leq 1.$$

(b) *The function $\psi : [0, 1] \to [0, 1 - \pi T(0)]$ with $\psi(F) = (1 - \pi)(1 - F) + \pi(1 - T(F))$ is piecewise linear, convex, strictly decreasing with $\psi(1) = 0$, and satisfies*

$$(1 - \pi)(1 - F) \leq \psi(F) \leq 1 - F \qquad \text{for all } F \in [0, 1].$$

(c) *The inverse function $\psi^{-1} : [0, 1 - \pi T(0)] \to [0, 1]$ is piecewise linear, convex, strictly decreasing with $\psi^{-1}(0) = 1$, and satisfies*

$$1 - \frac{a}{1 - \pi} \leq \psi^{-1}(a) \leq 1 - a \qquad \text{for all } a \in [0, 1 - \pi T(0)].$$

*Proof.* Going from a) to b) is simple: Since $\psi(F)$ is a convex combination of the piece-wise linear, strictly decreasing and convex functions $(1 - F)$ and $1 - T(F)$, it has the same properties. The bounds follow directly from the bounds on $T(F)$. The reverse follows from the same argument, again using that $\psi(F)$ and $T(F)$ are linear transformations of each other. Lastly, it is easy to see that $\psi(1) = 0$ is equivalent to $T(1) = 1$.

For the equivalence of b) and c), we first note that the inverse of a strictly decreasing piece-wise linear function always exists and is piece-wise linear. In addition, $\psi(1) = 0$ is clearly equivalent to $\psi^{-1}(1) = 0$. Next, we need that the inverse of any convex, decreasing function $f$ is convex and decreasing. We begin with the decreasing part: Consider $f(x) > f(y)$. Then since $f$ was decreasing

$$f^{-1}(f(x)) = x < y = f^{-1}(f(y)).$$

Now for convexity, we need to show that

$$f^{-1}(\lambda f(x) + (1 - \lambda)f(x)) \leq \lambda f^{-1}(f(x)) + (1 - \lambda)f^{-1}(f(y))$$

for any $\lambda \in [0, 1]$. We use the convexity of $f$ and that $f^{-1}$ is decreasing to calculate:

$$\begin{aligned} f^{-1}(\lambda f(x) + (1 - \lambda)f(x)) &\leq f^{-1}(f(\lambda x + (1 - \lambda)y)) \\ &= \lambda x + (1 - \lambda)y \\ &= f^{-1}(f(x)) + (1 - \lambda)f^{-1}(f(y)). \end{aligned}$$

It remains to show the equivalence of the bounds. For this, we note that the respective bounds for $\psi$ and $\psi^{-1}(F)$ are inverse of each other, so it is sufficient to show that for decreasing $f, g$, we have $f(x) \lesseqgtr g(x) \iff f^{-1}(y) \lesseqgtr g^{-1}(y)$. For this, consider $y = f(x)$. Then,

$$
\begin{aligned}
f^{-1}(y) \leq g^{-1}(y) &\iff f^{-1}(f(x)) \leq g^{-1}(f(x)) \\
&\iff g(f^{-1}(f(x))) \geq g(g^{-1}(f(x))) \\
&\iff g(x) \geq f(x)
\end{aligned}
$$

and analogous for $\geq$. $\qquad\square$

With this, we set $\tau := 1 - \pi T(0) \in [1 - \pi, 1]$ and extend $\psi^{-1}$ by zero to

$$
\hat{\psi}^{-1}(a) := \begin{cases} \psi^{-1}(a), & a \in [0, \tau], \\ 0, & a \in (\tau, 1]. \end{cases}
$$

By Proposition 7,

$$
\mathrm{ACC}_{\psi^{-1}}(g_N) = 1 - (1 - \pi)N(N - 1) \int_0^1 \hat{\psi}^{-1}(a)\, a^{N-2}\, da. \tag{13}
$$

The extension preserves convexity and monotonicity, with $\hat{\psi}^{-1}(0) = 1$, $\hat{\psi}^{-1}(1) = 0$, and the pointwise bounds

$$
\max\left\{0,\, 1 - \frac{a}{1-\pi}\right\} \leq \hat{\psi}^{-1}(a) \leq 1 - a \qquad (a \in [0, 1]). \tag{14}
$$

In particular, $\mathrm{ACC}_{\psi^{-1}}(g_N)$ is linear and ($L_1$)-continuous in $\hat{\psi}^{-1}$, such that whenever we can express

$$
\hat{\psi}^{-1}(a) = \sum_{i=0}^{\infty} w_i \hat{b}_{x_i}^{-1}(a)
$$

as the $L_1$-limit of sums of basis functions $\hat{b}_{x_i}^{-1}$, we get

$$
\mathrm{ACC}_{\psi^{-1}}(g_N) = \sum_{i=0}^{\infty} w_i \mathrm{ACC}_{\hat{b}_{x_i}^{-1}}(g_N).
$$

For the basis functions $\hat{b}_{x_i}^{-1}$, we consider the family of "hinge" functions:

**Definition 2** (Hinge family). *For $x \in (0, 1]$, define the hinge function as*

$$
\hat{b}_x^{-1}(a) := \begin{cases} 1 - \dfrac{a}{x}, & a \leq x, \\ 0, & a > x, \end{cases} \qquad a \in [0, 1]. \tag{15}
$$

With these, we have that

$$
\hat{b}_{1-\pi}^{-1}(a) = \max\left\{0, 1 - \frac{a}{1-\pi}\right\} \leq \psi^{-1}(a) \leq 1 - a = \hat{b}_1^{-1}(a). \tag{16}
$$

We now claim that $\hat{\psi}^{-1}$ fulfills the layed-out conditions if and only if it can be written as a convex combination $\hat{\psi}^{-1} = \sum_{i=0}^{\infty} w_i \hat{b}_{x_i}^{-1}$ with positive weights $w_i$ adding up to one, $x_i \leq 1$ and $\sum_{i=0}^{\infty} w_i \frac{1}{x_i} \leq \frac{1}{1-\pi}$:

**Lemma 7** (Characterization of $\hat{\psi}^{-1}$). *A function $\hat{\psi}^{-1} : [0, 1] \to [0, 1]$ satisfies the convexity, monotonicity, and slope constraints if and only if it can be written as*

$$
\hat{\psi}^{-1}(a) = \sum_{i=0}^{\infty} w_i\, \hat{b}_{x_i}^{-1}(a), \qquad a \in [0, 1],
$$

*where $\hat{b}_x^{-1}(a) := \max\{1 - a/x, 0\}$ are the hinge functions, and the parameters satisfy*

$$
w_i \geq 0, \qquad \sum_{i=0}^{\infty} w_i = 1, \qquad x_i \leq 1, \qquad \sum_{i=0}^{\infty} \frac{w_i}{x_i} \leq \frac{1}{1-\pi}.
$$

*Proof. (⇒:)* We begin by showing that any such convex combination gives us a valid $\hat{\psi}^{-1}$: Since the $\hat{b}_x^{-1}$ are piece-wise linear, decreasing, convex, and fulfill the boundary conditions, the same is true for any convex combination. Similarly, the $\hat{b}_x^{-1}$ are point-wise increasing in $x$, such that $\hat{b}_1^{-1}$ is an upper bound for any such convex combination. Lastly, any convex combination $\hat{\psi}^{-1} = \sum_{i=0}^{\infty} w_i \hat{b}_{x_i}^{-1}$ with $\sum_{i=0}^{\infty} w_i \frac{1}{x_i} \leq \frac{1}{1-\pi}$ has its derivative lower bounded by $-\frac{1}{1-\pi}$ while being a positive function, such that $\hat{\psi}^{-1}$ is lower bounded by $\hat{b}_{1-\pi}^{-1}$.

*(⇐:)* Now, we want to show that any valid $\hat{\psi}^{-1}$ can be written as such a convex combination. We thus fix any piece-wise linear, convex and strictly decreasing $\hat{\psi}^{-1}$ with $\psi^{-1}(0) = 1$, as well as

$$1 - \frac{a}{1-\pi} \leq \psi^{-1}(a) \leq 1 - a$$

for all $a \in [0, 1 - \pi T(0)]$.

We explicitly construct the sum representation as follows, for now ignoring the constraints: Let $(I_n)_{n \in \mathbb{N}}$ be an enumeration of the pieces of $\hat{\psi}^{-1}$, ordered from right to left. We build the sum inductively, matching the behavior of $\hat{\psi}^{-1}$ on the intervals $(I_n)_{n \leq k}$ at the kth step.

For the base case, we start with the empty sum, matching $\hat{\psi}^{-1}$ on the empty set.

Now for the induction case, we assume that

$$\hat{\psi}^{-1} = \sum_{i=0}^{k} w_i \hat{b}_{x_i}^{-1}$$

on the intervals $(I_k)_{k \leq n}$. We now simply pick $x_{k+1} = a_k$, where $a_k$ is the left endpoint of $I_k$ and pick

$$w_{k+1} = x_{k+1}\left(\alpha_{k+1} - \sum_{i=0}^{k} w_i \frac{1}{x_i}\right),$$

where $\alpha_{k+1}$ is the slope of $\hat{\psi}^{-1}$ on the interval $I_{k+1}$ such that

$$\alpha_{k+1} = -\sum_{i=0}^{k+1} w_i \frac{1}{x_i}$$

is indeed equal to the slope of the constructed sum $\hat{\psi}^{-1} = \sum_{i=0}^{k+1} w_i \hat{b}_{x_i}^{-1}$. Because $\hat{b}_{x_{k+1}}^{-1}(a) = 0$ for $a \geq x$, this does not change the sum's behavior on the previous intervals, so that the sum matches $\hat{\psi}^{-1}$ on all $I_n$ for $n$ up to $k+1$.

There thus exists sequences $w_i$ and $x_i$ such that

$$\hat{\psi}^{-1}(a) = \sum_{i=0}^{\infty} w_i \hat{b}_{x_i}^{-1},$$

where the series converges point-wise, and thus in $L_1$ via dominated convergence (because both the $\hat{b}^{-1}$ and $\hat{\psi}^{-1}$) are bounded.

We now claim that the constructed $w_k$ and $x_k$ fulfill our constraints: By construction, we have $x_k \leq 1$. Furthermore, because $\hat{\psi}^{-1}$ is convex and the slopes $\alpha_k$ thus increase (we go from right to left!), all $w_k$ are positive. Next, if $\sum_{i=0}^{\infty} w_i \neq 1$, the boundary condition $\hat{\psi}^{-1}(0) = 1$ would be violated. Lastly, for the sake of contradiction, assume that $\sum_{i=0}^{\infty} w_i \frac{1}{x_i} > \frac{1}{1-\pi}$. Then, we can find a finite $k$ such that $\sum_{i=0}^{k} w_i \frac{1}{x_i} > \frac{1}{1-\pi}$. There then exists an interval $I$ such that $\hat{b}_{x_i}^{-1}(a) = 1 - \frac{1}{x_i} a$ for all $a \in I$ and $i \leq k$. But that means that for $a \in I$

$$\hat{\psi}^{-1}(a) = \sum_{i=0}^{\infty} w_i \hat{b}_{x_i}^{-1}(a)$$

$$= \sum_{i=0}^{k} w_i \hat{b}_{x_i}^{-1}(a) + \sum_{i=k+1}^{\infty} w_i \hat{b}_{x_i}^{-1}(a)$$

$$\leq \sum_{i=0}^{k} w_i \hat{b}_{x_i}^{-1}(a) + \sum_{i=k+1}^{\infty} w_i$$

$$= \sum_{i=0}^{k} w_i(1 - \frac{a}{x_i}) + \sum_{i=k+1}^{\infty} w_i$$

$$= 1 - \sum_{i=0}^{k} w_i \frac{a}{x_i}$$

$$< 1 - \frac{a}{1 - \pi}$$

which contradicts Equation (16). $\qquad\square$

By Proposition 7, $\mathrm{ACC}(g_N^f)$ only depends on the function $\psi^{-1}$ induced by $f$. By Lemma 7, there are verifiers $f$ with concave ROC curves induced by any

$$\hat{\psi}^{-1}(a) = \sum_{i=1}^{\infty} w_i \, \hat{b}_{x_i}^{-1}(a) \quad \text{in } L^1([0,1]), \qquad w_i \geq 0, \ \sum_i w_i = 1, \ \sum_i \frac{w_i}{x_i} \leq \frac{1}{1-\pi}, \ x_i \in (0,1].$$

Now, using linearity, $L^1$-continuity, and slightly abusing notation by parameterizing the BoN accuracy as $\mathrm{ACC}(g_N^{\psi^{-1}})$, we get

$$\mathrm{ACC}(g_N^{\psi^{-1}}) = \sum_{i=1}^{\infty} w_i \, \mathrm{ACC}\left(g_N^{\hat{b}_{x_i}^{-1}}\right) = 1 - (1-\pi)N(N-1)\sum_{i=1}^{\infty} w_i \int_0^1 \hat{b}_{x_i}^{-1}(a) \, a^{N-2} \, da.$$

For each hinge $\hat{b}_x^{-1}(a) = \max\{1 - a/x, 0\}$ we can write

$$\int_0^1 \hat{b}_x^{-1}(a) \, a^{N-2} \, da = \int_0^x \left(1 - \frac{a}{x}\right) a^{N-2} \, da = \frac{x^{N-1}}{N(N-1)}.$$

Hence the accuracy simplifies to the closed form

$$\mathrm{ACC}(g_N^{\psi^{-1}}) = 1 - (1-\pi)\sum_{i=1}^{\infty} w_i \, x_i^{N-1}. \tag{17}$$

**Remark 1.** *The operator norm of $\mathcal{A}_N$ on $L^1([0,1])$ equals $(1-\pi)N(N-1)$; therefore, while the passage to the limit above is valid for each fixed $N$, the convergence need not be uniform in $N$.*

$\qquad\square$

### C.11 PROOF OF PROPOSITION 9

Consider any $\epsilon > 0$. Assume, that there is a convex combination of hinge functions $\tilde{\psi}^{-1} = \sum_{i=0}^{k} w_i \hat{b}_{x_i}^{-1}$ such that

$$||\hat{\psi}^{-1} - \tilde{\psi}^{-1}||_{L_1} \leq \frac{\epsilon}{2N(N+1)},$$

where $\hat{\psi}^{-1}$ is defined as in Appendix C.10.

Repeating the steps from Theorem 2, we get

$$\text{ACC}(g_{N+1}^{\tilde{\psi}^{-1}}) - \text{ACC}(g_N^{\tilde{\psi}^{-1}}) = (1 - \pi) \sum_{i=0}^{k} w_i(x_i^{N-1} - x_i^N) > 0$$

Since $\text{ACC}(g_N^{(\cdot)}) - \text{ACC}(g_{N+1}^{(\cdot)})$ is $2N(N+1)-$Lipschitz in the $L_1$ norm via the representation from Proposition 7, this means that $\text{ACC}(g_{N+1}^{\psi^{-1}}) + \epsilon \geq \text{ACC}(g_N^{\psi^{-1}})$. Because $\epsilon$ was arbitrary, we thus have

$$\text{ACC}_{\psi^{-1}}(g_{N+1}) \geq \text{ACC}_{\psi^{-1}}(g_N).$$

It remains to show that we can indeed $L_1$ approximate any $\hat{\psi}^{-1}$ by convex combinations of hinge functions. This is established by the following lemma:

**Lemma 8.** *The set of convex combinations of hinge functions is $L_1$-dense in the set of all decreasing, convex functions $\hat{\psi}^{-1}$ on $[0, 1]$ with the boundary conditions $\hat{\psi}^{-1}(0) = 1, \hat{\psi}^{-1}(1) = 0$.*

*Proof.* The key idea is that we can write piecewise linear decreasing convex functions as convex combinations of hinge functions (Step 1), and those are already dense in the space of convex functions (Step 2):

1. (Step 1): For any convex piece-wise linear function $\psi^{-1}$ with finitely many pieces and values in $[0, 1]$, we use the construction from the proof of Lemma 7 to explicitly write $\psi^{-1}$ as a convex combination of hinge functions. Because $\psi^{-1}$ has finitely many pieces, the resulting convex combination is finite.

2. (Step 2): Now, given any convex function $\hat{\psi}^{-1}$ with the boundary conditions, we can fix $n$ and evaluate $\hat{\psi}^{-1}$ on the points $K = \{\frac{k}{n} : k \in \mathbb{N} \wedge 0 \leq k \leq n\}$. Then the linear interpolator $\psi^{-1}$ of $\{a, \hat{\psi}^{-1}(a) : x \in K\}$ is a convex piece-wise linear function with finitely many pieces that fulfills the boundary conditions. In addition,

$$\int_0^1 |\hat{\psi}^{-1}(a) - \psi^{-1}(a)|da = \sum_{k=0}^{n-1} \int_{\frac{k}{n}}^{\frac{k+1}{n}} |\hat{\psi}^{-1}(a) - \psi^{-1}(a)|da$$
$$\leq \sum_{k=0}^{n-1} \frac{1}{n} \max_{a \in [\frac{k}{n}, \frac{k+1}{n}]} |\hat{\psi}^{-1}(a) - \psi^{-1}(a)|$$
$$\leq \sum_{k=0}^{n-1} \frac{2L}{n^2}$$
$$= \frac{2L}{n} \to 0,$$

where $L$ is the (uniform) Lipschitz constant of $f$, which exists for all convex real functions on compact intervals Lan (2020).

$\square$

## C.12 PROOF OF PROPOSITION 6

*Proof.* We first focus on the non-concave case: There, via Proposition 4, BoN accuracy is given by:

$$\text{ACC}(g_N^f) = 1 - (1 - \pi)N \int_0^1 \psi_f(F)^{N-1} \, dF,$$

where $\psi(F) = (1 - \pi)(1 - F) + \pi(1 - T_f(F))$ is the probability that a sample is rejected given FPR F for the verifier $f$.

For $\delta > 0$ consider $\tilde{f}$ with

$$T_{\tilde{f}}(F) = \begin{cases} T_f(F) & : F > \delta \\ 0 & : F \leq \delta \end{cases}.$$

Then by theorem 1, $\lim_{N\to\infty} \text{ACC}(g_N^{\tilde{f}}) = 0$.

But $\psi_f(F) = \psi_{\tilde{f}}$ for $F > \delta$, such that

$$|\text{ACC}(g_N^f) - \text{ACC}(g_N^{\tilde{f}})| = (1-\pi)N| \int_0^1 \psi_f(F)^{N-1}\, dF - \int_0^1 \psi_{\tilde{f}}(F)^{N-1}\, dF|$$

$$= (1-\pi)N| \int_0^1 \psi_f(F)^{N-1} - \psi_{\tilde{f}}(F)^{N-1}\, dF|$$

$$= (1-\pi)N| \int_0^\delta \psi_f(F)^{N-1} - \psi_{\tilde{f}}(F)^{N-1}\, dF|$$

$$= (1-\pi)N \int_0^\delta \psi_f(F)^{N-1}\, dF$$

$$\leq (1-\pi)N\delta$$

because $\psi_f$ is between zero and one. In particular, this is smaller than $\epsilon$ whenever $\delta < \frac{1}{(1-\pi)N}$. For a fixed upper bound on $N < n$, we can simply choose $\delta = \frac{1}{(1-\pi)n}$.

Now, consider $\tilde{f}'$ with

$$\mathbf{T}_{\tilde{f}'}(\mathbf{F}) = \begin{cases} \mathbf{T}_f(\mathbf{F}) & : \mathbf{F} > \delta \\ \mathbf{T}_f(\delta) & : \mathbf{F} \leq \delta \end{cases},$$

which leads to $\lim_{N\to\infty} \text{ACC}(g_N^{\tilde{f}'}) = 1$ according to the second case of Theorem 1.

In this case,

$$|\text{ACC}(g_N^f) - \text{ACC}(g_N^{\tilde{f}'})| = (1-\pi)N| \int_0^\delta \psi_f(F)^{N-1} - \psi_{\tilde{f}'}(F)^{N-1}\, dF|$$

$$= (1-\pi)N| \int_0^\delta \psi_f(F)^{N-1} - \psi_f(\delta)^{N-1}\, dF|$$

$$\leq (1-\pi)N2\delta$$

from where we proceed as before, this time setting $\delta = \frac{1}{2(1-\pi)n}$.

For the concave case, our argument has two steps: First, we show that there is a convex, decreasing piece-wise linear $\hat{\psi}^{-1}$ with $\hat{\psi}^{-1}(0) = 1$ and $\hat{\psi}^{-1}(1) = 0$ such that

$$|\text{ACC}(g_N^{\psi^{-1}}) - \text{ACC}(g_N^{\hat{\psi}^{-1}})| \leq \frac{\epsilon}{2}$$

for all $N \leq n$. Then, we show that for any piece-wise linear $\hat{\psi}^{-1}$, there is another convex, decreasing piece-wise linear $\tilde{\psi}^{-1}$ with $\tilde{\psi}^{-1}(0) = 1$ and $\tilde{\psi}^{-1}(1) = 0$ such that $\lim_{N\to\infty} \text{ACC}(g_N^{\tilde{\psi}^{-1}}) = 1$ and

$$|\text{ACC}(g_N^{\hat{\psi}^{-1}}) - \text{ACC}(g_N^{\tilde{\psi}^{-1}})| \leq \frac{\epsilon}{2}.$$

Then by Theorem 2, there is a verifier with concave ROC that induces $\tilde{\psi}^{-1}$, and by the triangle inequality that verifier meets the theorem statement.

The first step directly follows from Lemma 8 combined with the (uniform) $N(N-1)-$Lipschitzness in $L_1$ of $\text{ACC}_{(\cdot)}(g_N)$ for $N \leq n$. We note that the $\hat{\psi}^{-1}$ constructed that way fulfills the linear constraints from Equation (16), as a linear interpolation of the function $\psi^{-1}$ that fulfills the constraints.

For the second step using Theorem 2, we can write $\text{ACC}(g_N^{\hat{\psi}^{-1}}) = 1 - (1-\pi)\sum_{i=0}^\infty w_i \frac{1}{b_i}^{N-1}$, where $b_i = \frac{1}{x_i}$ and the $x_i$ fulfill the constraints from Theorem 2 because $\hat{\psi}^{-1}$ fulfilled Equation (16). Without loss of generality, we assume all $b_i \geq 1$ to be distinct as well as $b_1 = 1$ and $w_2 > 0$.

Then we have

$$c = \lim_{N \to \infty} \mathrm{ACC}(g_N^{\hat{\psi}^{-1}})$$

$$= \lim_{N \to \infty} 1 - (1 - \pi) \sum_{i=0}^{\infty} w_i \frac{1}{b_i}^{N-1}$$

$$= 1 - (1 - \pi) \sum_{i=0}^{\infty} w_i \lim_{N \to \infty} \frac{1}{b_i}^{N-1}$$

$$= 1 - (1 - \pi) w_1$$

where we used $\frac{1}{b_i}^{N-1} \to 0$ and applied dominated convergence to exchange the sum and limit, using that $w_i \frac{1}{b_i}^{N-1} \le w_i$ which sum up to one. This implies that $w_1 = \frac{1-c}{1-\pi} > 0$. Then for fixed $\epsilon > 0$ and $n \in \mathbb{N}$, there is a constant $\delta > 0$, such that $|\frac{1}{1+\delta}^{N-1} - 1| \le \frac{\epsilon}{4w_i(1-\pi)}$ and $|\frac{1}{b_2 - \frac{w_1}{w_2}\delta}^{N-1} - \frac{1}{b_2}^{N-1}| \le \frac{\epsilon}{4w_2(1-\pi)}$ for all $N \le n$. In particular, because $b_2 > 1$, we can find such $\delta$ in a way that guarantees $b_2 - \frac{w_2}{w_1}\delta > 1$.

With this, we define $\tilde{b}_1 = 1 + \delta$, $\tilde{b}_2 = b_2 - \frac{w_1}{w_2}\delta$ and $\tilde{b}_i = b_i$ for $i \ge 2$, as well as $\tilde{x}_i = \frac{1}{b_i}$ . With this,

$$\sum_{i=0}^{\infty} w_i (b_i - \tilde{b}_i) = -w_1 \delta + w_2 \frac{w_1}{w_2}\delta = 0,$$

such that $\tilde{b}_i$ fulfills the constraints from Theorem 2. This means there is a verifier $\tilde{f}$ with concave ROC such that $\mathrm{ACC}(g_N^{\tilde{\psi}^{-1}}) = 1 - (1 - \pi) \sum_{i=0}^{\infty} w_i \frac{1}{b_i}^{N-1}$.

Because all $\tilde{b}_i$ are strictly larger than one, we have

$$\lim_{N \to \infty} \mathrm{ACC}(g_N^{\tilde{f}}) = 1 - (1 - \pi) \sum_{i=0}^{\infty} w_i \lim_{N \to \infty} \frac{1}{\tilde{b}_i}^{N-1} = 1.$$

At the same time for $N < n$,

$$|\mathrm{ACC}(g_N^{\hat{\psi}^{-1}}) - \mathrm{ACC}(g_N^{\tilde{\psi}^{-1}})| \le |(1 - \pi) w_1 (\frac{1}{\delta+1}^{N} - 1)| + |(1 - \pi) w_2 \frac{1}{b_2 - \frac{w_1}{w_2}\delta}^{N-1} - \frac{1}{b_2}^{N-1}|$$

$$\le \frac{\epsilon}{4} + \frac{\epsilon}{4}$$

$$= \frac{\epsilon}{2}.$$

$\square$

## D  RELAXING ABSOLUTE CONTINUITY

Verbally prompting LLMs for risk score estimates has been shown to effectively produce calibrated and accurate scores (Cruz et al., 2024), but can also lead to limited score resolution (since in practice only a few discrete score values are generated by the model). However, many of our proofs require the score to be absolutely continuous. Despite this, empirically, our theoretical predictions remain valid across the board. In this section, we resolve this discrepancy, showing how our theory extends to discrete scores via a small modification to the definition of the ROC curve[2].

Assumption 1 requires the scores $f(x)$ to be continuous and have a density. However, in practice such as for our experiments in Section 6, the scores $f(x)$ are often discrete, taking on values in a finite set $S$. In this section, we show that a small modification of the ROC curve, which results in a smoothed version of $f$ which we call $\tilde{f}$, happens to coincide with how ROC curves are plotted in the popular sklearn package and makes our theoretical results work in the discrete case:

---

[2]This modification happens to coincide with the way the popular sklearn package plots ROC curves

**Definition 3** (Smoothed Score $\tilde{f}$). *Let $f(x)$ be a discrete scoring function that takes on finitely many values $s \in S$, and let $\epsilon > 0$ denote the smallest difference between any two distinct values in $S$. We define the smoothed score $\tilde{f}$ as:*

$$\tilde{f}(x_i) = f(x_i) + \frac{\epsilon}{2}\xi_i,$$

*where each $\xi_i \sim Unif[-1, 1]$ is an independent random variable. The resulting function $\tilde{f}(x)$ has a density.*

We will show that a) the ROC curve of $\tilde{f}$ is achieved by linearly interpolating the points

$$\left( \mathrm{F}, \mathrm{T}(\mathrm{F}) = \max \left\{ \mathrm{T}(g, h) : h \in \mathcal{H}(f), \ \mathrm{F}(g, h) = \mathrm{F} \right\} \right),$$

exactly as done in `sklearn`, that b) $\tilde{f}$ induces the same accuracy for BoN as $f$, and c) that the scaling of rejection sampling for $\tilde{f}$ is the same as the scaling for rejection sampling, allowing for mixtures of adjacent thresholds $\tau$.

**$\tilde{f}$ interpolates the ROC curve of $f$** We now show that the ROC curve induced by $\tilde{f}$ is a linear interpolation of the stepwise ROC curve defined by $f$. For discrete scores, the ROC curve

$$\max \left\{ \mathrm{T}(g, h) : h \in \mathcal{H}(f), \ \mathrm{F}(g, h) \leq \mathrm{F} \right\}$$

is a step function, since it only increases at values of F corresponding to thresholds $h^\tau$ where $\tau \in S$ is one of the finitely many values taken by $f(x)$.

First, we show that $\tilde{f}$ exactly recovers the endpoints of each step in the ROC curve induced by $f$. Suppose $f(x_1) < f(x_2)$ are two adjacent values in $S$. Setting the threshold $\tau = f(x_1) - \frac{\epsilon}{2}$ ensures that all $x$ with $f(x) \geq f(x_1)$ are accepted (with probability 1), while all others are rejected. This yields the same T/F point as thresholding at $\tau = f(x_1)$ on the original function $f$. Similarly, thresholding at $\tau = f(x_2) - \frac{\epsilon}{2}$ recovers the ROC point corresponding to $f(x_2)$. Hence, $\tilde{f}$ retains the same step endpoints as $f$.

Next, consider any threshold $\tau$ that lies strictly between $f(x_1)$ and $f(x_2)$, namely

$$\tau = f(x_1) + \left( q - \frac{1}{2} \right) \epsilon \quad \text{for some } q \in [0, 1].$$

This threshold induces the following acceptance behavior:

- any $x'$ such that $f(x') \geq f(x_2)$ will be accepted with probability one.
- $x_1$ will be accepted if and only if the corresponding noise $\xi$ exceeds $2q - 1$, which happens with probability $q$.

Thus, using this threshold for $\tilde{f}$ is equivalent to using the threshold $f(x_1)$ with probability $q$ and the threshold $f(x_2)$ with probability $1 - q$ for the original verifier $f$, which is equivalent to using a randomized threshold.

This random threshold behavior leads to the convex combination of ROC points, i.e., linearly interpolates between them Fawcett (2006).

**BoN Accuracy is unchanged with $\tilde{f}$** We condition on the event that $k$ out of $N$ samples $x$ achieve the (sample) maximum of the score $f(x)$. Then, BoN on $f$ picks one of these $x$ uniformly at random. Thus, we need to show that BoN on $\tilde{f}$ does the same.

By construction of the noise, only samples $x_i$ that maximize $f(x_i)$ can maximize $\tilde{f}(x_i)$ with nonzero probability. Among these, BoN on $\tilde{f}$ picks the one for which the noise variable $\xi_i$ is maximized. But because the $\xi_i$ are IID, this amounts to picking one of the $x_i$ maximizing $f(x_i)$, uniformly at random.

# E  ADDITIONAL EMPIRICAL RESULTS

This section provides additional empirical evidence to support our theoretical results, using different generator models, different verifier models, and different GSM8K test questions. These questions were chosen by running the `Qwen3-1.7B` generator on the whole GSM8K test set and selecting the first questions that were answered incorrectly.

**Implementation details.** Generator model responses are evaluated using the `lm-evaluation-harness` (Gao et al., 2024) package. Verifier models receive the task description, the test question, and the generator model's chain-of-thought and answer. For each generator response $x$, the verifier is prompted to reason over it step-by-step and to output a correctness *risk score* $f(x)$ at the end of its response. On MATH500, both generator and verifier models are capped to produce at most 5K output tokens for each answer. Model outputs are uncapped for GSM8K, as answers were generally much shorter, hence generation length was not an issue. In some cases, we sample multiple risk scores $f(x)$ for the same verifier and average them in order to obtain a less noisy score. Prompt templates and examples are shown in Appendix F.

For GSM8K, we evaluate $y(x)$ via exact match of the bracketed answer. As MATH500 often allows for multiple correct phrasings of the same answer, we use the `math-verify` package to parse answers and compare them to the ground truth in order to obtain $y(x)$. On GPQA, we evaluate $y(x)$ via exact match with the letter code representing the correct answer. Meanwhile, for humaneval, we evaluate using the test suite and environment from the `evalplus` package.

Due to an issue with the setting the random seed, our reported aggregate results on GSM8K, MATH500, GPQA and humaneval resampled the few-shot prompts once per question, while the results on individual GSM8K questions resampled them once per generation, potentially leading to more diversity in the responses to each question due to different few-shot examples for different generations to the same question.

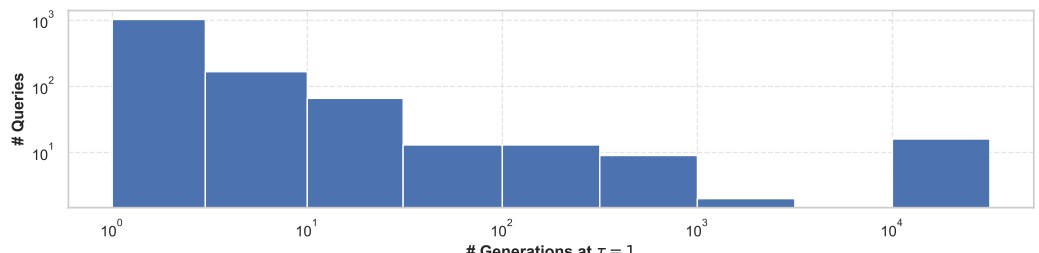

Figure E.1: Histogram of number of generations $x$ required to achieve $f(x) \geq \tau = 1$ across GSM8K items, capped at 10000. The vast majority of queries require less than ten samples, but 16 queries never meet the threshold. Verifier models: `Qwen3-32B` (blue) Generator: `Qwen3-1.7B`.

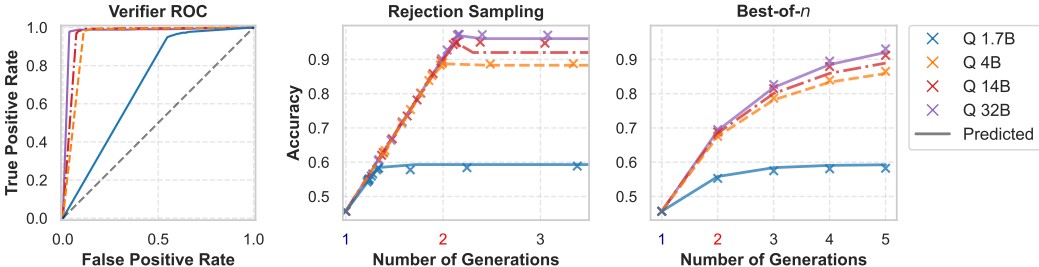

Figure E.2: Empirical performance (lines) of rejection sampling (middle) and BoN (right) on a GSM8K test question ($i = 2$), overlaid with predicted theoretical performance (x markers). Verification score obtained from a single chain of thought. Generator: `Qwen3-1.7B`.

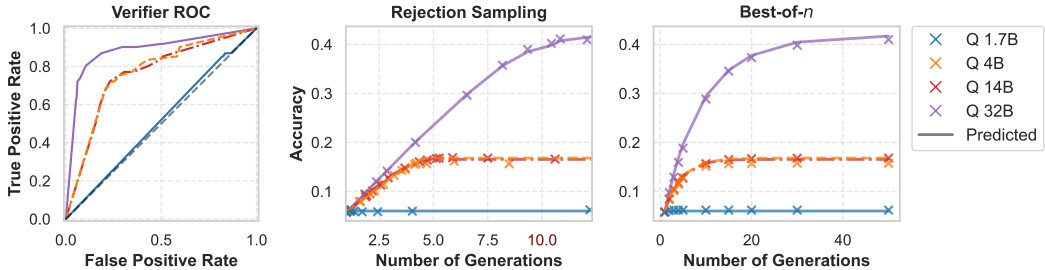

Figure E.3: Empirical performance (lines) of rejection sampling (middle) and BoN (right) on a GSM8K test question ($i = 7$), overlaid with predicted theoretical performance (× markers). Verification score obtained from a single chain of thought. Generator: `Qwen3-1.7B`.

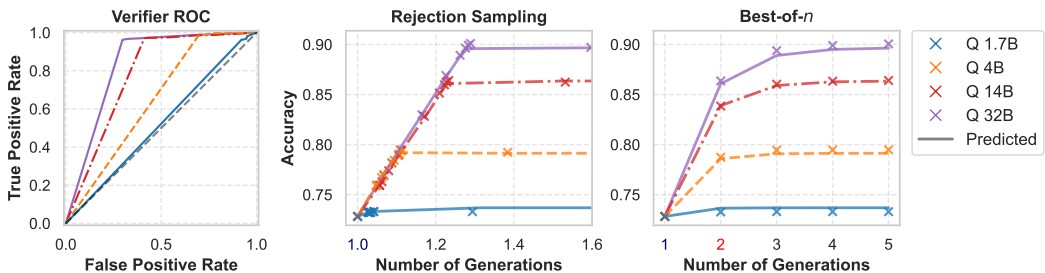

Figure E.4: Empirical performance (lines) of rejection sampling (middle) and BoN (right) on a GSM8K test question ($i = 7$), overlaid with predicted theoretical performance (× markers). Verification score obtained from a single chain of thought. Generator: `Qwen3-4B`.

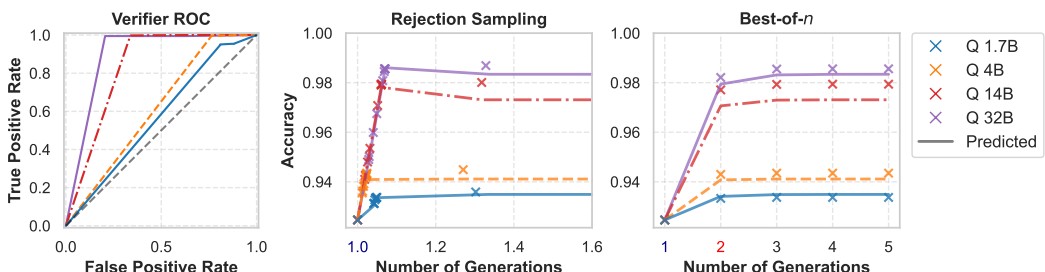

Figure E.5: Empirical performance (lines) of rejection sampling (middle) and BoN (right) on a GSM8K test question ($i = 8$), overlaid with predicted theoretical performance (× markers). Verification score obtained from a single chain of thought. Generator: `Qwen3-4B`.

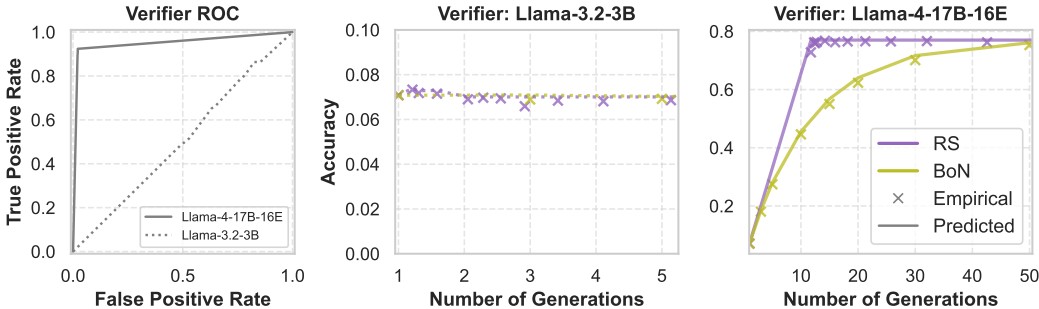

Figure E.6: A version of Fig. 3 on a different test question, $i = 7$. The same trend is observed: rejection sampling uses significantly less average compute than BoN for the same accuracy gain. Verification score obtained from a single chain of thought. Generator: `Llama-3.2-3B`, with $6.7\%$ generation accuracy.

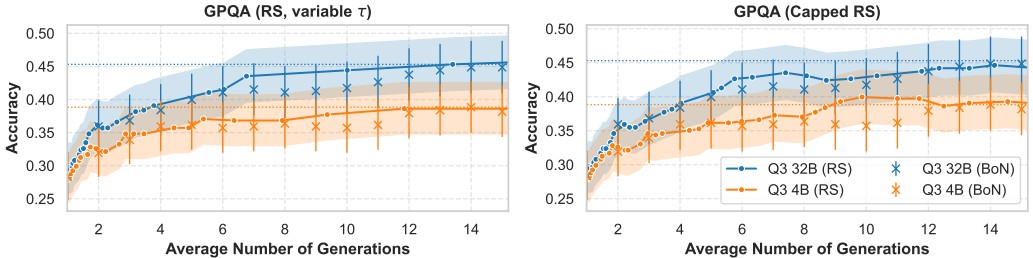

Figure E.7: Aggregate accuracy of RS with heuristic variable threshold (left) vs Capped RS (right) on GPQA (Rein et al., 2024). Dotted lines show Bo25 performance for the respective verifiers. While within error bar range, both RS with variable threshold and Capped RS appear to outperform BoN. Verifier models: `Qwen3-32B` (blue), `Qwen3-4B` (orange). Generator: `Qwen3-1.7B`. Error bars: Exact 90% CIs for accuracy.

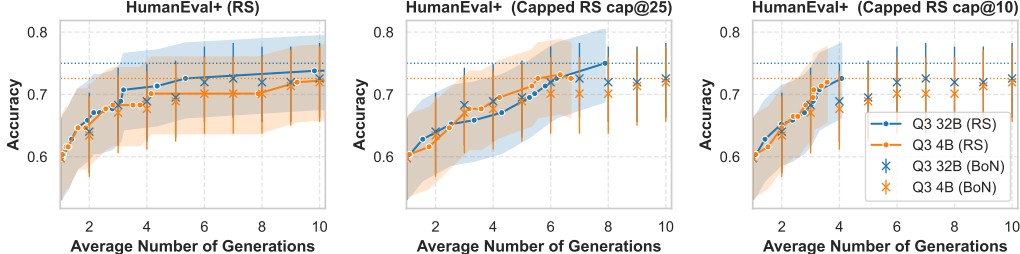

Figure E.8: Aggregate accuracy of RS with heuristic variable threshold (left) vs Capped RS (middle and right) on HumanEval+ (Liu et al., 2023) with different caps on the sample number. Dotted lines show Bo25 performance for the respective verifiers. While within error bar range, RS with variable threshold appeears to outperform BoN. Constant-threshold RS is sometimes outperformed by BoN when capped at 25 samples, but appears to consistently outperform BoN when capped at 10 samples. Verifier models: `Qwen3-32B` (blue), `Qwen3-4B` (orange). Generator: `Qwen3-1.7B`. Error bars: Exact 90% CIs for accuracy.

## F  GENERATION AND VERIFICATION PROMPTS

Listing 1 shows boiler-plate Python code used to encode GSM8K input document questions with few-shot examples. The output of this function will be tokenized and sent to the LLM generator model. The boxes that follow show the system prompt used for the LLM generator models, and task definition prompt used for the LLM verifier models. We use the HuggingFace `transformers` (Wolf et al., 2020) package for LLM completions. Note, that we use the same prompting strategy for Llama and Qwen3 models; in particular, we do *not* make use of Qwen3's "thinking" mode.

```python
def encode_with_chat_template(
    self: Generator,
    question: str,
    task: Task,
    num_fewshot: int,
) -> str:
    """Encode the input question and few-shot examples using a chat
        template."""
    conversation: list[dict] = []
    if self.system_prompt:
        conversation = [
            {"role": "system", "content": self.system_prompt}
        ]

    # Add fewshot examples
    for fewshot_example in task.fewshot_examples(num_fewshot, self.rng):

        # Add question as user message
        conversation.append({
            "role": "user",
            "content": f"Question: {fewshot_example['question']}",
        })

        # Add answer as assistant message
        conversation.append({
            "role": "assistant",
            "content": f"Answer: {fewshot_example['answer']}",
        })

    # Finally, add the task question as a user message
    conversation.append({
        "role": "user",
        "content": f"Question: {question}",
    })

    # Apply prompt template.
    # > add_generation_prompt=True will add the tokens to signal the
        start
    # > of the assistant's turn.
    prompt_up_to_assistant_turn = self.tokenizer.apply_chat_template(
        conversation=conversation,
        tokenize=False,
        add_generation_prompt=True,
    )

    # Append the desired prefix for the assistant's response
    encoded_prompt = prompt_up_to_assistant_turn + "Answer:"
    return encoded_prompt
```

Listing 1: Generator GSM8K chat-template encoding function, using the `transformers` package.

**Generator GSM8K System Prompt**

You are a helpful assistant that solves math problems. Think step by step. After reasoning, provide your answer in a separate line using the format: #### <answer>

**Generator MATH500 System Prompt**

You are a helpful assistant that solves math problems. Think step by step. After reasoning, provide your answer in a separate line  using the format: \\boxed{answer}

**Generator GPQA System Prompt**

You are a helpful assistant that solves STEM problems. Think step by step. After reasoning, provide your answer in a separate line  using the format: The correct answer is (insert answer here)

**Verifier Score Prompt**

You are a helpful assistant that verifies the correctness of a response to a question. Think step by step, carefully reasoning over each step of the response. Finally, your assessment should end with a score between 0 and 10 indicating how likely the response is to be correct. Your correctness score must strictly follow the format "[[<number>]]"; for example, "[[0]]" means the response is definitely incorrect and "[[10]]" means the response is definitely correct.

**Generator HumanEval+ System Prompt**

You are a helpful coding assistant. You will be shown a python function header and example behaviour. Please responsd with the completed function according to the specification. Keep all explanations constrained to comments within the code. Your answer should have the format
'''python

'''

**Verifier Score Prompt (Code)**

You are a helpful assistant that verifies the correctness of a python function. Think step by step, carefully reasoning over each line of code. Finally, your assessment should end with a score between 0 and 10 indicating how likely the code is to be correct. Your correctness score must strictly follow the format "[[<number>]]"; for example, "[[0]]" means the code is definitely incorrect and "[[10]]" means the code is definitely correct."""

