# OpenReview forum: "ROC-n-reroll: How verifier imperfection affects test-time scaling"
_ICLR.cc/2026/Conference — ICLR 2026 Poster_

### Official Review · Reviewer_WUvv · 2025-11-01

**Soundness:** 2
**Presentation:** 2
**Contribution:** 3
**Rating:** 6
**Confidence:** 2

**Summary:**

This paper provides a theoretical and empirical analysis of how verifier imperfections influence test-time scaling methods such as Best-of-N (BoN) and Rejection Sampling (RS). It shows that the accuracy of these methods is determined by the geometry of the verifier’s ROC curve, rather than by specific implementation details. It proves that RS outperforms BoN for fixed compute budgets and that both converge to the same accuracy as compute increases. The authors validate these findings empirically using Qwen and LLaMA models on GSM8K and MATH500, confirming that high-compute performance cannot be extrapolated from low-compute behavior.

**Strengths:**

1. The paper provides a theoretical understanding for resampling-based inference-time compute scaling, a timely and important topic in the era.
2. The analysis connects verifier imperfection to scaling behavior via the ROC curve. The link between ROC geometry and compute-performance scaling provides a unifying lens to reason about imperfect verifiers.
3. The paper is well-motivated, and the structure is clear.

**Weaknesses:**

1. The experiments only focus on two mathematical reasoning datasets (GSM8K and MATH500), where numeric ground truths simplify verification. I wonder whether the theoretical insights generalize to other reasoning datasets (e.g., commonsense reasoning) or open-ended generation (e.g., summarization, dialogue) or code generation.
2. Although the paper briefly discusses connections to RL (Appendix C.1), it doesn’t empirically test and validate these hypotheses. A comparison to small-scale RL fine-tuning (e.g., DPO, PPO) would strengthen the generalization of this argument.
3. The empirical performance curves in Figure 1 are only shown for a hand-picked example (test question 58).
4. The paper provides no actionable method to improve verifiers or scaling strategies — it only diagnoses why scaling may fail. Especially, one of the paper's findings in the abstract is that it is "generally impossible to predict the high-compute performance" from low-compute observations, which is a negative result without constructive guidance on how to solve this problem.

**Questions:**

See in weaknesses.

---

> ### Author Response · Authors · 2025-11-20
>
> Thank you for the positive review and valuable feedback! We have added additional experiments on GPQA to the revised manuscript and address your comments below:
>
> - "The experiments only focus on two mathematical reasoning datasets (GSM8K and MATH500), where numeric ground truths simplify verification. I wonder whether the theoretical insights generalize to other reasoning datasets (e.g., commonsense reasoning) or open-ended generation (e.g., summarization, dialogue) or code generation."
>    - Our theoretical results do not require the ground truth to be easily evaluable, but this is required to properly evaluate empirical results at scale. We have added results for GPQA (STEM reasoning, evaluated in a multiple choice fashion) to the appendix. While these results are noisier than the ones on MATH and GSM8K, Rejection Sampling again slightly outperforms Best-of-N, as suggested by our theory.
> - “The empirical performance curves in Figure 1 are only shown for a hand-picked example (test question 58).”
>    - Our theoretical results precisely characterize single-question performance, therefore we show an example of theoretical predictions vs empirical results for single-question performance - for one question in the main text due to space constraints and numerous others in the appendix. We show aggregate results for the whole GSM8K and MATH500 benchmark in Figure 4, and also added such results for GPQA in the appendix for the rebuttal revision.
> - “The paper provides no actionable method to improve verifiers or scaling strategies”
>    - We disagree. Our empirical results demonstrate that incorporating elements of rejection sampling, where possible, rather than fully relying on best-of-N can improve test-time scaling.
> - “One of the paper's findings [...] is a negative result without constructive guidance on how to solve this problem.”
>    - We respectfully disagree about this being a weakness. Our theoretical result shows that without stronger assumptions, the problem of predicting test-time high-compute performance from low-compute observations *is not solvable*, and our empirical results show that such assumptions do not seem to hold in practice. As such, we view our result as clear practical guidance to not rely on these kinds of predictions.

---

### Official Review · Reviewer_NcCh · 2025-11-01

**Soundness:** 4
**Presentation:** 3
**Contribution:** 3
**Rating:** 8
**Confidence:** 2

**Summary:**

This work investigates the performance of two LLM test-time scaling methods in scenarios with an imperfect verifier:
- Rejection Sampling (RS): generates responses sequentially until one passes verification.
- Best-of-N (BoN): generates N responses in parallel and selects the one with the highest verification score.

Within a formal statistical framework, the authors rigorously show that the accuracy of either method for solving a single query under a fixed compute budget is determined precisely by the ROC curve corresponding to the generator and verifier.

High-level takeaways from the analysis include:
1. RS outperforms BoN under fixed compute (assuming the ROC curve is concave), although both converge to the same accuracy when compute is unlimited.
2. For either method, high-compute accuracy cannot be reliably predicted from low-compute performance.

Theoretical findings are validated through experiments, lending further support to the results.

**Strengths:**

This paper presents a solid and well-executed contribution to the LLM research community, even though it does not introduce new algorithms. The theoretical analysis of RS and BoN is thorough and rigorous, providing fresh insights into their behaviors when given *imperfect verifiers*. The work offers practical takeaways for researchers and practitioners considering repeated sampling approaches.

*(Disclaimer: Some technical content extends beyond my core expertise, and thus I have not verified the correctness of all proofs in detail.)*

**Weaknesses:**

I don't see major weaknesses in this work, assuming that the theoretical analyses are correct.


Some suggestions:

- The takeaway message "RS outperforms BoN under fixed compute" requires the assumption that the ROC curve (for the particular query under consideration) is concave, according to Proposition 5. I think this assumption should be stated more explicitly in the abstract and introduction, otherwise the takeaway message alone could be slightly misleading.
In reality, there is no guarantee *a priori* that the ROC curve should be concave.

- The proof for BoN's limit accuracy in Theorem 1 (which equals that of RS in Proposition 2) is quite complicated.
On the other hand, we might think of (or approximate) BoN in the infinite-compute limit as (1) generating many samples, (2) keeping the samples whose scores exceed $1 - \delta$ (with $\delta \rightarrow 0$), (3) breaking ties and picking one of the high-score samples randomly.
In this case, BoN's accuracy is effectively the posterior probability that a generated sample is correct, conditioned on its score exceeding $1 - \delta$.
This probability can be easily derived by the Bayes rule, and doesn't fundamentally differ from the analysis for RS.
I wonder how far from rigor this analysis is, and whether it could offer some intuitions about Theorem 1 and the takeaway message that BoN and RS reach the same accuracy with infinite samples.

- Use a notation other than $\pi$ (Line 137) to avoid confusion with the mathematical constant $\pi \approx 3.14$.

- Line 373, "Comparing with Proposition 1": should be "Proposition 2".

- Include and discuss more related works, which could situate this paper more clearly in the literature on LLM test-time scaling.
Examples include [1, 2, 3] that provide theoretical analyses for different test-time scaling methods, and [4] that also investigates the limits of LLM repeated sampling when given imperfect verifiers (albeit more empirically).

[1] Are More LLM Calls All You Need? Towards Scaling Laws of Compound Inference Systems (NeurIPS 2024)

[2] Provable Scaling Laws for the Test-Time Compute of Large Language Models (NeurIPS 2025)

[3] When More is Less: Understanding Chain-of-Thought Length in LLMs (ICLR 2025 Workshop)

[4] Inference Scaling fLaws: The Limits of LLM Resampling with Imperfect Verifiers (arXiv 2024)

**Questions:**

See "weaknesses" above.

---

> ### Author Response · Authors · 2025-11-20
>
> Thank you for the positive review! We have updated the paper to incorporate most of your suggestions:
> - “Line 373, "Comparing with Proposition 1": should be "Proposition 2".”
>     - Good catch! We fixed this in the updated paper.
> - “Include and discuss more related works”
>    - Thank you for pointing out these papers. We incorporated them into the related work sections in the main text and the appendix, respectively.
> - “The takeaway message "RS outperforms BoN under fixed compute" requires the assumption that the ROC curve is concave [...] this assumption should be stated more explicitly in the abstract and introduction..”
>    - Thank you for pointing this out. We added additional qualifiers to the respective statements.
> - “The proof for BoN's limit accuracy in Theorem 1 (which equals that of RS in Proposition 2) is quite complicated”
>    - Thanks for sharing this possible alternative way to prove Theorem 1. We agree with your intuition, and think that this argument can be made rigorous when the distribution of verification scores is discrete and bounded: Then, as $N$ goes to infinity, any BoN sample will include the maximal verification score $f^\{\star}$ with high probability. Conditioned on this, the BoN Accuracy simply equals the accuracy of rejection sampling with the threshold $\tau = f^\{\star}$. However, we believe it would be nontrivial to turn this approach into a rigorous proof, when f is continuous. In particular, it seems difficult to concretely pick $\delta$ as a function of $N$ in a way that ensures that BoN’s accuracy indeed matches/approximates the accuracy of the base distribution conditional on $f(x)>1-\delta$.

---

### Official Review · Reviewer_RwZD · 2025-11-02

**Soundness:** 3
**Presentation:** 3
**Contribution:** 2
**Rating:** 6
**Confidence:** 2

**Summary:**

This paper studies rejection sampling and best-of-N sampling for LLMs with imperfect verifiers. The core idea is that, for a fixed sampling budget, the performance of both methods is determined by the verifier's ROC curve and the generator's base accuracy. The authors show this theoretically and provide empirical validation on math datasets.

**Strengths:**

1. Interesting problem, clean formalization, and clear scope that focuses on RS and BoN.
2. The result that RS outperforms BoN with fixed average compute seems useful in practice.
3. Empirical experiments show consistent results with theory's prediction.

**Weaknesses:**

1. **Limited evaluation domain**: Can the authors also consider other benchmark categories such as coding or more general QA? This is critical for evaluation whether the conclusion generalizes.
2. **Compute metric mismatch and missing hybrid method**
	- The compute metric for RS is defined as an expectation, while the compute metric for BoN is defined as the deterministic N. The stopping time of  RS creates variance but the analysis optimize only the mean. Also, the experiments on MATH500 and GSM8K has a cap at 25 (line 448), which changes the RS distribution in a way not covered by the theory.
	- Taking a step further, it feels natural to consider a hybrid approach where BoN early-stops if a certain score threshold is reached (or RS early stops if a certain N is reached). Although the authors mention this as future work, I believe that this is a baseline much more realistic than RS without stopping.

**Questions:**

- Can you verify the results on more diverse benchmarks?
- How will a hybrid method change your conclusions? Will this hybrid method outperform both RS and BoN? Could you test it empirically?

---

> ### Author Response · Authors · 2025-11-20
>
> Thank you for the positive review! In response to your review, we have added more detailed experiments on hybrid methods as well as on the GPQA benchmark to the updated paper. You can find more details below:
>
>  - "Limited evaluation domain: Can the authors also consider other benchmark categories such as coding or more general QA? This is critical for evaluation whether the conclusion generalizes. Can you verify the results on more diverse benchmarks?"
>    - We have added results for GPQA (STEM reasoning, evaluated in a multiple choice fashion) to the appendix (see page 39). While these results are a lot noisier than the ones on MATH and GSM8K, Rejection Sampling/Hybrid methods seem to slightly outperform Best-of-N.
> - “The stopping time of RS creates variance but the analysis optimize only the mean.”
>    - We agree. This is one of the reasons why we repeatedly mention that RS is less practical than BoN (Lines 297, 482, 102). As the variance of a geometric distribution with parameter $p$ is given by $\frac{1-p}{p^2}$, it would not be difficult to incorporate something akin to a variance-penalty into per-query analysis of RS. That said, we still believe that expected compute is the most suitable “common currency” to compare both methods, especially for large-scale deployments where variance averages out.
>  - “The experiments on MATH500 and GSM8K has a cap at 25 (line 448), which changes the RS distribution in a way not covered by the theory.”
>    - Yes, we originally restricted the RS distribution to save computational resources and deal with the mentioned impracticalities of RS. In new more comprehensive experiments, we found that for large thresholds, rejection sampling with a constant large threshold takes prohibitively long to run for a small subset of hard questions in GSM8K (around 1%). This is because our generator consistently fails to answer these questions correctly, which leads to consistently low verifier scores. As large thresholds are necessary for most other questions to achieve high accuracy, we added experiments using full rejection sampling with difficulty-dependent thresholds $\tau(q)$: For “easy” questions, where the generator’s accuracy exceeds 1%, we use a threshold $\tau = t \leq 1$, while for hard questions, we use a more relaxed threshold of $\tau = t/2$.  In Section 5, we added a detailed description of the new experiments and clarified and extended the discussion on our previous capped RS results.
>  - “It feels natural to consider a hybrid approach where BoN early-stops if a certain score threshold is reached (or RS early stops if a certain N is reached)”. How will a hybrid method change your conclusions? Will this hybrid method outperform both RS and BoN? Could you test it empirically?
>    - You are correct that the capping the number of RS samples at 25 in our initial experiments changed the RS distribution. By returning the highest-scoring of the 25 samples rather than an incorrect “null output” as in the original experiment, what we did essentially becomes the hybrid method you suggested. We have replaced the original experiments in Figure 4 with this method, and indeed find that it outperforms BoN, while not requiring difficulty-dependent thresholds.
>    - While it is generally difficult to analyze these hybrid methods, we can provide guarantees for a special case: When the score $f(x)$ is upper bounded by one, early stopping BoN whenever we find a sample with $f(x)=1$ will use at most as much compute than BoN, but have the same (expected) accuracy.

---

### Official Review · Reviewer_dsXu · 2025-11-04

**Soundness:** 3
**Presentation:** 2
**Contribution:** 3
**Rating:** 6
**Confidence:** 4

**Summary:**

This paper presents a timely study on how imperfections in a verifier affect the performance of test-time scaling, where the goal is to improve accuracy by spending more compute at inference time. The authors analyze two commonly used techniques: rejection sampling (RS) and Best-of-N (BoN). They show that the performance of both methods is fully determined by the geometry of the verifier’s ROC curve. For RS, they prove that performance in the low-compute and high-compute regimes depends on the local geometry of the ROC curve — specifically, how the ROC behaves near the operating points corresponding to large and small false-positive rates (FPR). In the high-compute regime, accuracy is governed by the slope of the ROC near FPR $\to$ 0. For BoN, performance depends on global ROC properties.

Their theoretical results, supported by experiments, show that RS is more compute-efficient than BoN in the finite compute regime, while both methods converge to the same asymptotic accuracy as compute goes to infinity. A key insight is that test-time scaling cannot be extrapolated: low-compute performance does not predict high-compute performance unless one knows the ROC geometry.

**Strengths:**

- One of the main strengths of this paper is that the authors present a simple and interpretable mathematical framework for analyzing how verifier imperfections impact the effectiveness of test-time scaling. Their formulation leverages classical concepts from statistical learning theory (e.g., ROC curves, true/false positive rates), providing a clean lens through which to reason about the verifier. This contribution is valuable because it connects modern LLM sampling practices with well-established theoretical tools, which may inspire clearer definitions and analysis in future research.

- Another strength is that the theoretical predictions align well with empirical results. For example, in Figure 3, the experiments on LLaMA models show the expected behavior: RS outperforms BoN under finite compute budgets, consistent with the theoretical claim that RS is more compute-efficient.

- Overall, the paper offers practical guidance on how to deploy test-time scaling methods when compute is constrained. It clarifies when additional sampling is worthwhile, and when verifier limitations make further compute ineffective.

**Weaknesses:**

- One weakness of this paper is its clarity / organization. I think the clarity can be significantly improved by including a table of notations in the main manuscript. On the first pass, it was difficult to follow the derivations because I had to look back to recall the notation.

- The theory models correctness as a binary variable (either correct or incorrect). This abstraction simplifies analysis, but it may limit applicability in settings where output quality is continuous or graded, e.g., in RL where partial correctness matters. It is unclear how the theoretical results extend to cases with non-binary reward structures or scalar score models, which are common in modern LLM evaluation and verification frameworks.

**Questions:**

Here are both questions and comments:

- This question might be a bit more abstract: the weakness stated above leads me to wonder whether compute could be defined differently in this setting. In the paper, compute is measured as the total number of generations. However, one could instead measure compute as the total number of tokens generated. If compute were defined in terms of token usage rather than completions, the relative advantage of RS could change. In particular, BoN may appear more favorable under a token-based compute budget, since it generates a fixed number of completions of comparable length, whereas RS may generate highly variable amounts of text before accepting a sample. Have the authors thought about these directions, if this is not sound I am happy to discuss.

- There is an iid assumption on the samples generated for BoN; is this pivotal in the proofs? I'm not sure this can be met in practice so it leaves me wondering if this is a drawback at all, or it doesn't matter and is just a definition.

-  I guess $\tau$ should be in [0, 1] in line 141

- It feels confusing to me to say "let the score $f$ to be fixed"; this sounds like the score is fixed and not the verifier. Perhaps calling it the verifier as done in line 301 is better.

- Does the theory depend at all if $h_F$ is not unique, i.e., there are multiple solutions to Equation (1)?

---

> ### Author Response · Authors · 2025-11-20
>
> Thank you for the positive review! We have updated the paper and reply to each of your points in order, below:
>
> - “One weakness of this paper is its clarity/organization.“
>    - Thank you for the feedback in terms of clarity! We have moved the notation table above the references. If you found specific aspects of the notation confusing beyond that, please let us know so we can improve them.
> - “The theory models correctness as a binary variable. [...] may limit applicability in settings where output quality is continuous”
>    - We do agree that, while out of this work’s scope, extending the results to non-binary rewards would be valuable. However, we think that this would be quite challenging due to the lack of an abstraction for the joint distribution between the verifier and general rewards that is as effective as the ROC curve. That said, to the best of our knowledge, modern LLM evaluation still overwhelmingly relies on binary scores. For example, all six of the headline results reported for the recent Kimi K2 Thinking model [1] are on benchmarks with a binary ground truth reward. As such, we believe that understanding the binary-reward setting is highly valuable on its own.
> - Regarding the questions:
>   - “If compute were defined in terms of token usage rather than completions, the relative advantage of RS could change.”
>     - This is an interesting point. Perhaps surprisingly, we found that (in expectation) the advantage of RS compared to BoN is not affected by considering token usage rather than the number of samples: Let $T_i$ be the (IID) number of tokens in the ith sample, and $K$ the number of samples used by RS. Then, because $T_i$ does not depend on the Event ${K\geq i}$, Wald’s theorem [2] yields that the expected number of tokens used by RS equals $E[K]*E[T]$. As the expected number of tokens for BoN is just $N * E[T]$, the advantage of Rejection sampling is preserved in this setting. We have added a footnote on this in the updated paper (Page 4).
>    - “There is an iid assumption on the samples generated for BoN; is this pivotal in the proofs?”
>      - The IID assumption is important for the proofs. In practice, it is simple to achieve by using fixed generation parameters, varying the random seed.
>   - “I guess tau should be in [0, 1] in line 141”
>     - Good catch! We rewrote this to increase clarity.
>   - “It feels confusing to me to say "let the score to be fixed"; this sounds like the score is fixed and not the verifier. Perhaps calling it the verifier as done in line 301 is better.”
>      - Thank you for pointing this out. The updated version now consistently refers to the verifier f when talking about the function and the (verifier) score $f(x)$ when talking about function values evaluated at a specific $x$.
>   - "Does the theory depend at all if HF is not unique, i.e., there are multiple solutions to Equation (1)?”
>      - This is not a problem since all maximizers have the same True/False positive rate, and we show that RS accuracy is fully determined by True/False positive rates.
>
> [1] https://moonshotai.github.io/Kimi-K2/thinking.html
>
> [2]https://courses.csail.mit.edu/6.042/past-devel/archive/spring00/archive/handouts/H46-wald.pdf

---

### Author Response · Authors · 2025-12-03

We thank the reviewers for their valuable and insightful feedback. We are glad they found or work well-motivated ($\textcolor{blue}{WUvv}$), providing fresh insights ($\textcolor{orange}{NcCh}$) into an interesting ($\textcolor{purple}{RwZD}$), timely and important ($\textcolor{blue}{WUvv}$) problem. We are encouraged that they found our formalization to be a clean ($\textcolor{purple}{RwZD}$), simple and interpretable ($\textcolor{red}{dsXu}$) unifying lens to reason about imperfect verifiers ($\textcolor{blue}{WUvv}$). We are also pleased that reviewers found our theoretical analysis thorough and rigorous ($\textcolor{orange}{NcCh}$), while aligning well with our empirical results ($\textcolor{red}{dsXu}$,$\textcolor{purple}{RwZD}$) and being useful for practice ($\textcolor{red}{dsXu}$,$\textcolor{purple}{RwZD}$,$\textcolor{orange}{NcCh}$).

In response to the reviewers’ feedback, we have updated the paper as follows:
   - In response to reviewer $\textcolor{red}{dsXu}$ and $\textcolor{orange}{NcCh}$’s comments, we have improved clarity by qualifying statements in the introduction more precisely, more clearly distinguishing between verifiers and scores, as well as moving our notation table to the main text. We have also included additional material in our related work section.
   - To answer reviewer $\textcolor{red}{dsXu}$’s question, we have added additional analysis on measuring compute in terms of tokens rather than number of generated texts to Appendix B.
   - Responding to reviewer $\textcolor{purple}{RwZD}$’s comments, we have updated our discussion on aggregate experiments. We now cleanly distinguish between “full” rejection sampling and “hybrid methods” that combine rejection sampling with a fallback to best-of-N to prevent excessive sampling for particularly hard questions.
   - Furthermore, in response to reviewers $\textcolor{blue}{WUvv}$ and $\textcolor{purple}{RwZD}$ call for additional evaluation domains, we have added additional experiments on GPQA (STEM reasoning) and HumanEval+ (Coding) to Appendix E.

---

### Meta-Review · Area_Chair_NoCg · 2026-01-02

**Summary:**

This paper provides a theoretical analysis of two test-time scaling techniques, namely rejection sampling and Best-of-N. All reviewers recommend acceptance and recognize the paper's theoretical contributions. The main initial concerns focus on limited empirical verification, questions about the generality of some arguments, the alignment between the theoretical model and more realistic settings, and several writing or clarity issues. In the revision, the authors added new experimental results and clarified the presentation. As a result, the original concerns have largely been addressed, and any remaining discussion points (e.g., how to translate the theory into more directly actionable methods) are not substantial enough to affect the acceptance decision.

**Reviewer Concerns:**

Reviewer RwZD and Reviewer WUvv request additional empirical verification, and the authors added new results to address this point. Reviewer dsXu also raise concerns about the paper's clarity and organization, which the revision has notably improved.

Some minor issues remain. For example, Reviewer dsXu asks whether the results can be extended beyond binary rewards, and the authors acknowledge this limitation and provide a reasonable explanation of why such an extension is non-trivial. Reviewer WUvv also asks for clearer practical implications, and the authors argue that the negative result itself gives actionable guidance, which I agree with. Overall, these remaining points do not affect the acceptance decision.

**Reviewer Scores:**

Reviewer NcCh has already assigned a score of 8 and state that the paper has no weaknesses, so the evaluation is likely final.

Reviewer dsXu and Reviewer RwZD may increase their scores, given that the authors have addressed their requests by improving the writing and adding additional empirical results.

It is less clear whether Reviewer WUvv will adjust their score upward, as it depends on whether the concern about actionable methods remains. However, it is very unlikely that this reviewer will lower the score.

---

### Decision · Program_Chairs · 2026-01-26

Accept (Poster)